# Finite-Time Analysis for Double Q-learning

Huaqing Xiong[1], Lin Zhao[2], Yingbin Liang[1], and Wei Zhang[*3,4]

[1]The Ohio State University
[2]National University of Singapore
[3]Southern University of Science and Technology
[4]Peng Cheng Laboratory
[1]{xiong.309, liang.889}@osu.edu;  [2]elezhli@nus.edu.sg;  [3,4]zhangw3@sustech.edu.cn

## Abstract

Although Q-learning is one of the most successful algorithms for finding the best action-value function (and thus the optimal policy) in reinforcement learning, its implementation often suffers from large overestimation of Q-function values incurred by random sampling. The double Q-learning algorithm proposed in Hasselt (2010) overcomes such an overestimation issue by randomly switching the update between two Q-estimators, and has thus gained significant popularity in practice. However, the theoretical understanding of double Q-learning is rather limited. So far only the asymptotic convergence has been established, which does not characterize how fast the algorithm converges. In this paper, we provide the first non-asymptotic (i.e., finite-time) analysis for double Q-learning. We show that both synchronous and asynchronous double Q-learning are guaranteed to converge to an $\epsilon$-accurate neighborhood of the global optimum by taking $\tilde{\Omega}\left( \left( \frac{1}{(1-\gamma)^6 \epsilon^2} \right)^{\frac{1}{\omega}} + \left( \frac{1}{1-\gamma} \right)^{\frac{1}{1-\omega}} \right)$ iterations, where $\omega \in (0, 1)$ is the decay parameter of the learning rate, and $\gamma$ is the discount factor. Our analysis develops novel techniques to derive finite-time bounds on the difference between two inter-connected stochastic processes, which is new to the literature of stochastic approximation.

## 1 Introduction

Q-learning is one of the most successful classes of reinforcement learning (RL) algorithms, which aims at finding the optimal action-value function or Q-function (and thus the associated optimal policy) via off-policy data samples. The Q-learning algorithm was first proposed by Watkins and Dayan (1992), and since then, it has been widely used in various applications including robotics (Tai and Liu, 2016), autonomous driving (Okuyama et al., 2018), video games (Mnih et al., 2015), to name a few. Theoretical performance of Q-learning has also been intensively explored. The asymptotic convergence has been established in Tsitsiklis (1994); Jaakkola et al. (1994); Borkar and Meyn (2000); Melo (2001); Lee and He (2019). The non-asymptotic (i.e., finite-time) convergence rate of Q-learning was firstly obtained in Szepesvári (1998), and has been further studied in (Even-Dar and Mansour, 2003; Shah and Xie, 2018; Wainwright, 2019; Beck and Srikant, 2012; Chen et al., 2020) for synchronous Q-learning and in (Even-Dar and Mansour, 2003; Qu and Wierman, 2020) for asynchoronous Q-learning.

One major weakness of Q-learning arises in practice due to the large overestimation of the action-value function (Hasselt, 2010; Hasselt et al., 2016). Practical implementation of Q-learning involves using the maximum *sampled* Q-function to estimate the maximum *expected* Q-function (where the

---

[*]Corresponding author

expectation is taken over the randomness of reward). Such an estimation often yields a large positive bias error (Hasselt, 2010), and causes Q-learning to perform rather poorly. To address this issue, double Q-learning was proposed in Hasselt (2010), which keeps two Q-estimators (i.e., estimators for Q-functions), one for estimating the maximum Q-function value and the other one for update, and continuously changes the roles of the two Q-estimators in a random manner. It was shown in Hasselt (2010) that such an algorithm effectively overcomes the overestimation issue of the vanilla Q-learning. In Hasselt et al. (2016), double Q-learning was further demonstrated to substantially improve the performance of Q-learning with deep neural networks (DQNs) for playing Atari 2600 games. It inspired many variants (Zhang et al., 2017; Abed-alguni and Ottom, 2018), received a lot of applications (Zhang et al., 2018a,b), and have become one of the most common techniques for applying Q-learning type of algorithms (Hessel et al., 2018).

Despite its tremendous empirical success and popularity in practice, theoretical understanding of double Q-learning is rather limited. Only the asymptotic convergence was provided in Hasselt (2010); Weng et al. (2020c). There has been no non-asymptotic result on how fast double Q-learning converges. From the technical standpoint, such finite-time analysis for double Q-learning does not follow readily from those for the vanilla Q-learning, because it involves two randomly updated Q-estimators, and the coupling between these two random paths significantly complicates the analysis. This goes much more beyond the existing techniques for analyzing the vanilla Q-learning that handles the random update of a single Q-estimator. Thus, *the goal of this paper is to develop new finite-time analysis techniques that handle the inter-connected two random path updates in double Q-learning and provide the convergence rate.*

## 1.1 Our contributions

The main contribution of this paper lies in providing the first finite-time analysis for double Q-learning with both the synchronous and asynchronous implementations.

- We show that synchronous double Q-learning with a learning rate $\alpha_t = 1/t^\omega$ (where $\omega \in (0,1)$) attains an $\epsilon$-accurate global optimum with at least the probability of $1 - \delta$ by taking $\Omega\left(\left(\frac{1}{(1-\gamma)^6\epsilon^2}\ln\frac{|\mathcal{S}||\mathcal{A}|}{(1-\gamma)^7\epsilon^2\delta}\right)^{\frac{1}{\omega}} + \left(\frac{1}{1-\gamma}\ln\frac{1}{(1-\gamma)^2\epsilon}\right)^{\frac{1}{1-\omega}}\right)$ iterations, where $\gamma \in (0,1)$ is the discount factor, $|\mathcal{S}|$ and $|\mathcal{A}|$ are the sizes of the state space and action space, respectively.

- We further show that under the same accuracy and high probability requirements, asynchronous double Q-learning takes $\Omega\left(\left(\frac{L^4}{(1-\gamma)^6\epsilon^2}\ln\frac{|\mathcal{S}||\mathcal{A}|L^4}{(1-\gamma)^7\epsilon^2\delta}\right)^{\frac{1}{\omega}} + \left(\frac{L^2}{1-\gamma}\ln\frac{1}{(1-\gamma)^2\epsilon}\right)^{\frac{1}{1-\omega}}\right)$ iterations, where $L$ is the covering number specified by the exploration strategy.

Our results corroborate the design goal of double Q-learning, which opts for better accuracy by making less aggressive progress during the execution in order to avoid overestimation. Specifically, our results imply that in the high accuracy regime, double Q-learning achieves the same convergence rate as vanilla Q-learning in terms of the order-level dependence on $\epsilon$, which further indicates that the high accuracy design of double Q-learning dominates the less aggressive progress in such a regime. In the low-accuracy regime, which is not what double Q-learning is designed for, the cautious progress of double Q-learning yields a slightly weaker convergence rate than Q-learning in terms of the dependence on $1 - \gamma$.

From the technical standpoint, our proof develops new techniques beyond the existing finite-time analysis of the vanilla Q-learning with a single random iteration path. More specifically, we model the double Q-learning algorithm as two alternating stochastic approximation (SA) problems, where one SA captures the error propagation between the two Q-estimators, and the other captures the error dynamics between the Q-estimator and the global optimum. For the first SA, we develop new techniques to provide the finite-time bounds on the two inter-related stochastic iterations of Q-functions. Then we develop new tools to bound the convergence of Bernoulli-controlled stochastic iterations of the second SA conditioned on the first SA.

## 1.2 Related work

Due to the rapidly growing literature on Q-learning, we review only the theoretical results that are highly relevant to our work.

Q-learning was first proposed in Watkins and Dayan (1992) under finite state-action space. Its asymptotic convergence has been established in Tsitsiklis (1994); Jaakkola et al. (1994); Borkar and Meyn (2000); Melo (2001) through studying various general SA algorithms that include Q-learning as a special case. Along this line, Lee and He (2019) characterized Q-learning as a switched linear system and applied the results of Borkar and Meyn (2000) to show the asymptotic convergence, which was also extended to other Q-learning variants. Another line of research focuses on the finite-time analysis of Q-learning which can capture the convergence rate. Such non-asymptotic results were firstly obtained in Szepesvári (1998). A more comprehensive work (Even-Dar and Mansour, 2003) provided finite-time results for both synchronous and asynchoronous Q-learning. Both Szepesvári (1998) and Even-Dar and Mansour (2003) showed that with linear learning rates, the convergence rate of Q-learning can be exponentially slow as a function of $\frac{1}{1-\gamma}$. To handle this, the so-called rescaled linear learning rate was introduced to avoid such an exponential dependence in synchronous Q-learning (Wainwright, 2019; Chen et al., 2020) and asynchronous Q-learning (Qu and Wierman, 2020). The finite-time convergence of Q-learning was also analyzed with constant step sizes (Beck and Srikant, 2012; Chen et al., 2020; Li et al., 2020). Moreover, the polynomial learning rate, which is also the focus of this work, was investigated for both synchronous (Even-Dar and Mansour, 2003; Wainwright, 2019) and asynchronous Q-learning (Even-Dar and Mansour, 2003). In addition, it is worth mentioning that Shah and Xie (2018) applied the nearest neighbor approach to handle MDPs on infinite state space.

Differently from the above extensive studies of vanilla Q-learning, theoretical understanding of double Q-learning is limited. The only asymptotic convergence guarantee was provided by Hasselt (2010). A concurrent work (Weng et al., 2020c) studied the mean-square errors for double Q-learning given the assumption that it converges to a unique fixed point. Both of the abovementioned papers do not provide the non-asymptotic (i.e., finite-time) analysis on how fast double Q-learning converges. This paper provides the first finite-time analysis for double Q-learning.

The vanilla Q-learning algorithm has also been studied for the function approximation case, i.e., the Q-function is approximated by a class of parameterized functions. In contrast to the tabular case, even with linear function approximation, Q-learning has been shown not to converge in general (Baird, 1995). Strong assumptions are typically imposed to guarantee the convergence of Q-learning with function approximation (Bertsekas and Tsitsiklis, 1996; Zou et al., 2019; Chen et al., 2019; Du et al., 2019; Xu and Gu, 2019; Cai et al., 2019; Weng et al., 2020a,b). Regarding double Q-learning, it is still an open topic on how to design double Q-learning algorithms under function approximation and under what conditions they have theoretically guaranteed convergence.

## 2 Preliminaries on Q-learning and Double Q-learning

In this section, we introduce the Q-learning and the double Q-learning algorithms.

### 2.1 Q-learning

We consider a $\gamma$-discounted Markov decision process (MDP) with a finite state space $\mathcal{S}$ and a finite action space $\mathcal{A}$. The transition probability of the MDP is given by $P : \mathcal{S} \times \mathcal{A} \times \mathcal{S} \to [0, 1]$, that is, $\mathbb{P}(\cdot|s, a)$ denotes the probability distribution of the next state given the current state $s$ and action $a$. We consider a random reward function $R_t$ at time $t$ drawn from a fixed distribution $\phi : \mathcal{S} \times \mathcal{A} \times \mathcal{S} \mapsto \mathbb{R}$, where $\mathbb{E}\{R_t(s, a, s')\} = R_{sa}^{s'}$ and $s'$ denotes the next state starting from $(s, a)$. In addition, we assume $|R_t| \leq R_{\max}$. A policy $\pi := \pi(\cdot|s)$ characterizes the conditional probability distribution over the action space $\mathcal{A}$ given each state $s \in \mathcal{S}$.

The action-value function (i.e., Q-function) $Q^\pi \in \mathbb{R}^{|\mathcal{S}| \times |\mathcal{A}|}$ for a given policy $\pi$ is defined as

$$
\begin{aligned}
Q^\pi(s, a) :=& \mathbb{E}\left[\sum_{t=0}^\infty \gamma^t R_t(s, \pi(s), s') \Big| s_0 = s, a_0 = a\right] \\
=& \mathbb{E}_{\substack{s' \sim P(\cdot|s,a) \\ a' \sim \pi(\cdot|s')}}\left[R_{sa}^{s'} + \gamma Q^\pi(s', a')\right],
\end{aligned}
\tag{1}
$$

where $\gamma \in (0, 1)$ is the discount factor. Q-learning aims to find the Q-function of an optimal policy $\pi^*$ that maximizes the accumulated reward. The existence of such a $\pi^*$ has been proved in the classical

MDP theory ([Bertsekas and Tsitsiklis, 1996](#)). The corresponding optimal Q-function, denoted as $Q^*$, is known as the unique fixed point of the Bellman operator $\mathcal{T}$ given by

$$\mathcal{T}Q(s,a) = \mathbb{E}_{s' \sim P(\cdot|s,a)} \left[ R_{sa}^{s'} + \gamma \max_{a' \in U(s')} Q(s',a') \right], \tag{2}$$

where $U(s') \subset \mathcal{A}$ is the admissible set of actions at state $s'$. It can be shown that the Bellman operator $\mathcal{T}$ is $\gamma$-contractive in the supremum norm $\|Q\| := \max_{s,a} |Q(s,a)|$, i.e., it satisfies

$$\|\mathcal{T}Q - \mathcal{T}Q'\| \leq \gamma \|Q - Q'\|. \tag{3}$$

The goal of Q-learning is to find $Q^*$, which further yields $\pi^*(s) = \arg\max_{a \in U(s)} Q^*(s,a)$. In practice, however, exact evaluation of the Bellman operator [(2)](#) is usually infeasible due to the lack of knowledge of the transition kernel of MDP and the randomness of the reward. Instead, Q-learning draws random samples to estimate the Bellman operator and iteratively learns $Q^*$ as

$$Q_{t+1}(s,a) = (1 - \alpha_t(s,a))Q_t(s,a) + \alpha_t(s,a)\left(R_t(s,a,s') + \gamma \max_{a' \in U(s')} Q_t(s',a')\right), \tag{4}$$

where $R_t$ is the sampled reward, $s'$ is sampled by the transition probability given $(s,a)$, and $\alpha_t(s,a) \in (0,1]$ denotes the learning rate.

## 2.2 Double Q-learning

Although Q-learning is a commonly used RL algorithm to find the optimal policy, it can suffer from overestimation in practice ([Smith and Winkler, 2006](#)). To overcome this issue, [Hasselt (2010)](#) proposed double Q-learning given in Algorithm [1](#).

---

**Algorithm 1** Synchronous Double Q-learning ([Hasselt, 2010](#))

---

1: **Input:** Initial $Q_1^A, Q_1^B$.
2: **for** $t = 1, 2, \ldots, T$ **do**
3:     Assign learning rate $\alpha_t$.
4:     Randomly choose either UPDATE(A) or UPDATE(B) with probability 0.5, respectively.
5:     **for** each $(s,a)$ **do**
6:         observe $s' \sim P(\cdot|s,a)$, and sample $R_t(s,a,s')$.
7:         **if** UPDATE(A) **then**
8:            Obtain $a^* = \arg\max_{a'} Q_t^A(s',a')$
9:            $Q_{t+1}^A(s,a) = Q_t^A(s,a) + \alpha_t(s,a)(R_t(s,a,s') + \gamma Q_t^B(s',a^*) - Q_t^A(s,a))$
10:        **else if** UPDATE(B) **then**
11:           Obtain $b^* = \arg\max_{b'} Q_t^B(s',b')$
12:           $Q_{t+1}^B(s,a) = Q_t^B(s,a) + \alpha_t(s,a)(R_t(s,a,s') + \gamma Q_t^A(s',b^*) - Q_t^B(s,a))$
13:        **end if**
14:     **end for**
15: **end for**
16: **Output:** $Q_T^A$ (or $Q_T^B$).

---

Double Q-learning maintains two Q-estimators (i.e., Q-tables): $Q^A$ and $Q^B$. At each iteration of Algorithm [1](#), one Q-table is randomly chosen to be updated. Then this chosen Q-table generates a greedy optimal action, and the other Q-table is used for estimating the corresponding Bellman operator for updating the chosen table. Specifically, if $Q^A$ is chosen to be updated, we use $Q^A$ to obtain the optimal action $a^*$ and then estimate the corresponding Bellman operator using $Q^B$. As shown in [Hasselt (2010)](#), $\mathbb{E}[Q^B(s',a^*)]$ is likely smaller than $\max_a \mathbb{E}[Q^A(s',a)]$, where the expectation is taken over the randomness of the reward for the same state-action pair. In this way, such a two-estimator framework of double Q-learning can effectively reduce the overestimation.

**Synchronous and asynchronous double Q-learning:** In this paper, we study the finite-time convergence rate of double Q-learning in two different settings: synchronous and asynchronous implementations. For synchronous double Q-learning (as shown in Algorithm [1](#)), all the state-action pairs of the chosen Q-estimator are visited simultaneously at each iteration. For the asynchronous case, only one state-action pair is updated in the chosen Q-table. Specifically, in the latter case, we sample a

trajectory $\{(s_t, a_t, R_t, i_t)\}_{t=0}^{\infty}$ under a certain exploration strategy, where $i_t \in \{A, B\}$ denotes the index of the chosen Q-table at time $t$. Then the two Q-tables are updated based on the following rule:

$$Q_{t+1}^i(s, a)$$
$$= \begin{cases} Q_t^i(s, a), & (s, a) \neq (s_t, a_t) \text{ or } i \neq i_t; \\ (1 - \alpha_t(s, a))Q_t^i(s, a) + \alpha_t(s, a)\Big(R_t(s, a, s') + \gamma Q_t^{i^c}\big(s', \underset{a' \in U(s')}{\arg\max} Q_t^i(s', a')\big)\Big), & \text{otherwise,} \end{cases}$$

where $i^c = \{A, B\} \setminus i$.

We next provide the boundedness property of the Q-estimators and the errors in the following lemma, which is typically necessary for the finite-time analysis.

**Lemma 1.** *For either synchronous or asynchronous double Q-learning, let $Q_t^i(s, a)$ be the value of either Q table corresponding to a state-action pair $(s, a)$ at iteration $t$. Suppose $\|Q_0^i\| \leq \frac{R_{\max}}{1-\gamma}$. Then we have $\|Q_t^i\| \leq \frac{R_{\max}}{1-\gamma}$ and $\|Q_t^i - Q^*\| \leq V_{\max}$ for all $t \geq 0$, where $V_{\max} := \frac{2R_{\max}}{1-\gamma}$.*

Lemma 1 can be proved by induction arguments using the triangle inequality and the uniform boundedness of the reward function, which is seen in Appendix B.

# 3   Main results

We present our finite-time analysis for the synchronous and asynchronous double Q-learning in this section, followed by a sketch of the proof for the synchronous case which captures our main techniques. The detailed proofs of all the results are provided in the Supplementary Materials.

## 3.1   Synchronous double Q-learning

Since the update of the two Q-estimators is symmetric, we can characterize the convergence rate of either Q-estimator, e.g., $Q^A$, to the global optimum $Q^*$. To this end, we first derive two important properties of double Q-learning that are crucial to our finite-time convergence analysis.

The first property captures the stochastic error $\|Q_t^B - Q_t^A\|$ between the two Q-estimators. Since double Q-learning updates alternatingly between these two estimators, such an error process must decay to zero in order for double Q-learning to converge. Furthermore, how fast such an error converges determines the overall convergence rate of double Q-learning. The following proposition (which is an informal restatement of Proposition 1 in Appendix C.1) shows that such an error process can be *block-wisely* bounded by an exponentially decreasing sequence $G_q = (1 - \xi)^q V_{\max}$ for $q = 0, 1, 2, \ldots$, and some $\xi \in (0, 1)$. Conceptually, as illustrated in Figure 1, such an error process is upper-bounded by the blue-colored piece-wise linear curve.

**Proposition 1.** *(Informal) Consider synchronous double Q-learning under a polynomial learning rate $\alpha_t = \frac{1}{t^\omega}$ with $\omega \in (0, 1)$. We divide the time horizon into blocks $[\hat{\tau}_q, \hat{\tau}_{q+1})$ for $q \geq 0$, where $\hat{\tau}_0 = 0$ and $\hat{\tau}_{q+1} = \hat{\tau}_q + c_1 \hat{\tau}_q^\omega$ with some $c_1 > 0$. Fix $\hat{\epsilon} > 0$. Then for any $n$ such that $G_n \geq \hat{\epsilon}$ and under certain conditions on $\hat{\tau}_1$ (see Appendix C.1), we have*

$$\mathbb{P}\left[\forall q \in [0, n], \forall t \in [\hat{\tau}_{q+1}, \hat{\tau}_{q+2}), \|Q_t^B - Q_t^A\| \leq G_{q+1}\right] \geq 1 - c_2 n \exp\left(-\frac{c_3 \hat{\tau}_1^\omega \hat{\epsilon}^2}{V_{\max}^2}\right),$$

*where the positive constants $c_2$ and $c_3$ are specified in Appendix C.1.*

Proposition 1 implies that the two Q-estimators approach each other asymptotically, but does not necessarily imply that they converge to the optimal action-value function $Q^*$. Then the next proposition (which is an informal restatement of Proposition 2 in Appendix C.2) shows that as long as the high probability event in Proposition 1 holds, the error process $\|Q_t^A - Q^*\|$ between either Q-estimator (say $Q^A$) and the optimal Q-function can be *block-wisely* bounded by an exponentially decreasing sequence $D_k = (1 - \beta)^k \frac{V_{\max}}{\sigma}$ for $k = 0, 1, 2, \ldots$, and $\beta \in (0, 1)$. Conceptually, as illustrated in Figure 1, such an error process is upper-bounded by the yellow-colored piece-wise linear curve.

**Proposition 2.** *(Informal) Consider synchronous double Q-learning using a polynomial learning rate $\alpha_t = \frac{1}{t^\omega}$ with $\omega \in (0, 1)$. We divide the time horizon into blocks $[\tau_k, \tau_{k+1})$ for $k \geq 0$, where*

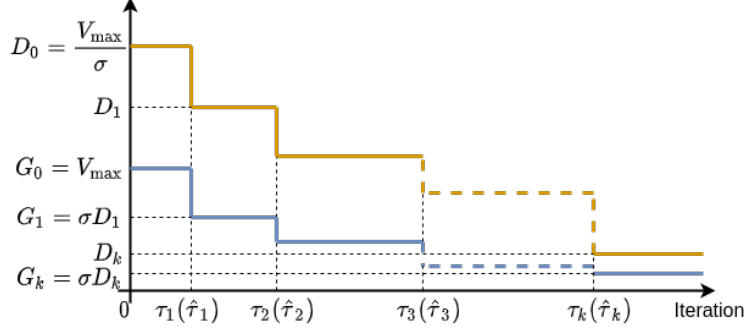

Figure 1: Illustration of sequence $\{G_k\}_{k \geq 0}$ as a block-wise upper bound on $\left\|Q_t^B - Q_t^A\right\|$, and sequence $\{D_k\}_{k \geq 0}$ as a block-wise upper bound on $\left\|Q_t^A - Q^*\right\|$ conditioned on the first upper bound event. See Appendix A for a numerical example to illustrate the block-wise upper bounds.

$\tau_0 = 0$ and $\tau_{k+1} = \tau_k + c_4 \tau_k^\omega$ with some $c_4 > 0$. Fix $\tilde{\epsilon} > 0$. Then for any $m$ such that $D_m \geq \tilde{\epsilon}$ and under certain conditions on $\tau_1$ (see Appendix C.2), we have

$$\mathbb{P}\left[\forall k \in [0, m], \forall t \in [\tau_{k+1}, \tau_{k+2}), \left\|Q_t^A - Q^*\right\| \leq D_{k+1}|E, F\right] \geq 1 - c_5 m \exp\left(-\frac{c_6 \tau_1^\omega \tilde{\epsilon}^2}{V_{\max}^2}\right),$$

where $E$ and $F$ denote certain events defined in (12) and (13) in Appendix C.2, and the positive constants $c_4, c_5,$ and $c_6$ are specified Appendix C.2.

As illustrated in Figure 1, the two block sequences $\{\hat{\tau}_q\}_{q \geq 0}$ in Proposition 1 and $\{\tau_q\}_{q \geq 0}$ in Proposition 2 can be chosen to coincide with each other. Then combining the above two properties followed by further mathematical arguments yields the following main theorem that characterizes the convergence rate of double Q-learning. We will provide a proof sketch for Theorem 1 in Section 3.3, which explains the main steps to obtain the supporting properties of Proposition 1 and 2 and how they further yield the main theorem.

**Theorem 1.** *Fix $\epsilon > 0$ and $\gamma \in (1/3, 1)$. Consider synchronous double Q-learning using a polynomial learning rate $\alpha_t = \frac{1}{t^\omega}$ with $\omega \in (0, 1)$. Let $Q_T^A(s, a)$ be the value of $Q^A$ for a state-action pair $(s, a)$ at time $T$. Then we have $\mathbb{P}(\left\|Q_T^A - Q^*\right\| \leq \epsilon) \geq 1 - \delta$, given that*

$$T = \Omega\left(\left(\frac{V_{\max}^2}{(1-\gamma)^4 \epsilon^2} \ln \frac{|\mathcal{S}||\mathcal{A}|V_{\max}^2}{(1-\gamma)^5 \epsilon^2 \delta}\right)^{\frac{1}{\omega}} + \left(\frac{1}{1-\gamma} \ln \frac{V_{\max}}{(1-\gamma)\epsilon}\right)^{\frac{1}{1-\omega}}\right), \tag{5}$$

*where $V_{\max} = \frac{2R_{\max}}{1-\gamma}$.*

Theorem 1 provides the finite-time convergence guarantee in high probability sense for synchronous double Q-learning. Specifically, double Q-learning attains an $\epsilon$-accurate optimal Q-function with high probability with at most $\Omega\left(\left(\frac{1}{(1-\gamma)^6 \epsilon^2} \ln \frac{1}{(1-\gamma)^7 \epsilon^2}\right)^{\frac{1}{\omega}} + \left(\frac{1}{1-\gamma} \ln \frac{1}{(1-\gamma)^2 \epsilon}\right)^{\frac{1}{1-\omega}}\right)$ iterations. Such a result can be further understood by considering the following two regimes. In the high accuracy regime, in which $\epsilon \ll 1 - \gamma$, the dependence on $\epsilon$ dominates, and the time complexity is given by $\Omega\left(\left(\frac{1}{\epsilon^2} \ln \frac{1}{\epsilon^2}\right)^{\frac{1}{\omega}} + \left(\ln \frac{1}{\epsilon}\right)^{\frac{1}{1-\omega}}\right)$, which is optimized as $\omega$ approaches to 1. In the low accuracy regime, in which $\epsilon \gg 1 - \gamma$, the dependence on $\frac{1}{1-\gamma}$ dominates, and the time complexity can be optimized at $\omega = \frac{6}{7}$, which yields $T = \tilde{\Omega}\left(\frac{1}{(1-\gamma)^7 \epsilon^{7/3}} + \frac{1}{(1-\gamma)^7}\right) = \tilde{\Omega}\left(\frac{1}{(1-\gamma)^7 \epsilon^{7/3}}\right)$.

Furthermore, Theorem 1 corroborates the design effectiveness of double Q-learning, which overcomes the overestimation issue and hence achieves better accuracy by making less aggressive progress in each update. Specifically, comparison of Theorem 1 with the time complexity bounds of vanilla synchronous Q-learning under a polynomial learning rate in Even-Dar and Mansour (2003) and Wainwright (2019) indicates that in the high accuracy regime, double Q-learning achieves the same convergence rate as vanilla Q-learning in terms of the order-level dependence on $\epsilon$. Clearly, the design of double Q-learning for high accuracy dominates the performance. In the low-accuracy regime

(which is not what double Q-learning is designed for), double Q-learning achieves a slightly weaker convergence rate than vanilla Q-learning in Even-Dar and Mansour (2003); Wainwright (2019) in terms of the dependence on $1 - \gamma$, because its nature of less aggressive progress dominates the performance.

## 3.2 Asynchronous Double Q-learning

In this subsection, we study the asynchronous double Q-learning and provide its finite-time convergence result.

Differently from synchronous double Q-learning, in which all state-action pairs are visited for each update of the chosen Q-estimator, asynchronous double Q-learning visits only one state-action pair for each update of the chosen Q-estimator. Therefore, we make the following standard assumption on the exploration strategy (Even-Dar and Mansour, 2003):

**Assumption 1.** *(Covering number) There exists a covering number $L$, such that in consecutive $L$ updates of either $Q^A$ or $Q^B$ estimator, all the state-action pairs of the chosen Q-estimator are visited at least once.*

The above conditions on the exploration are usually necessary for the finite-time analysis of asynchronous Q-learning. The same assumption has been taken in Even-Dar and Mansour (2003). Qu and Wierman (2020) proposed a mixing time condition which is in the same spirit.

Assumption 1 essentially requires the sampling strategy to have good visitation coverage over all state-action pairs. Specifically, Assumption 1 guarantees that consecutive $L$ updates of $Q^A$ visit each state-action pair of $Q^A$ at least once, and the same holds for $Q^B$. Since $2L$ iterations of asynchronous double Q-learning must make at least $L$ updates for either $Q^A$ or $Q^B$, Assumption 1 further implies that any state-action pair $(s, a)$ must be visited at least once during $2L$ iterations of the algorithm. In fact, our analysis allows certain relaxation of Assumption 1 by only requiring each state-action pair to be visited during an interval with a certain probability. In such a case, we can also derive a finite-time bound by additionally dealing with a conditional probability.

Next, we provide the finite-time result for asynchronous double Q-learning in the following theorem.

**Theorem 2.** *Fix $\epsilon > 0, \gamma \in (1/3, 1)$. Consider asynchronous double Q-learning under a polynomial learning rate $\alpha_t = \frac{1}{t^\omega}$ with $\omega \in (0, 1)$. Suppose Assumption 1 holds. Let $Q_T^A(s, a)$ be the value of $Q^A$ for a state-action pair $(s, a)$ at time $T$. Then we have $\mathbb{P}(\|Q_T^A - Q^*\| \leq \epsilon) \geq 1 - \delta$, given that*

$$T = \Omega \left( \left( \frac{L^4 V_{\max}^2}{(1-\gamma)^4 \epsilon^2} \ln \frac{|\mathcal{S}||\mathcal{A}| L^4 V_{\max}^2}{(1-\gamma)^5 \epsilon^2 \delta} \right)^{\frac{1}{\omega}} + \left( \frac{L^2}{1-\gamma} \ln \frac{\gamma V_{\max}}{(1-\gamma)\epsilon} \right)^{\frac{1}{1-\omega}} \right). \quad (6)$$

Comparison of Theorem 1 and 2 indicates that the finite-time result of asynchronous double Q-learning matches that of synchronous double Q-learning in the order dependence on $\frac{1}{1-\gamma}$ and $\frac{1}{\epsilon}$. The difference lies in the extra dependence on the covering time $L$ in Theorem 2. Since synchronous double Q-learning visits all state-action pairs (i.e., takes $|\mathcal{S}||\mathcal{A}|$ sample updates) at each iteration, whereas asynchronous double Q-learning visits only one state-action pair (i.e., takes only one sample update) at each iteration, a more reasonable comparison between the two should be in terms of the overall sample complexity. In this sense, synchronous and asynchronous double Q-learning algorithms have the sample complexities of $|\mathcal{S}||\mathcal{A}|T$ (where $T$ is given in (5)) and $T$ (where $T$ is given in (6)), respectively. Since in general $L \gg |\mathcal{S}||\mathcal{A}|$, synchronous double-Q is more efficient than asynchronous double-Q in terms of the overall sampling complexity.

## 3.3 Proof Sketch of Theorem 1

In this subsection, we provide an outline of the technical proof of Theorem 1 and summarize the key ideas behind the proof. The detailed proof can be found in Appendix C.

In general, although we adopt the epoch-wise analysis technique which has been used for studying vanilla Q-learning in Bertsekas and Tsitsiklis (1996); Even-Dar and Mansour (2003), the analysis of double Q-learning requires to handle two interconnected SAs, so that the analysis of single SA in Even-Dar and Mansour (2003) cannot be directly applied. Thus, we need to develop new tools to handle such a new challenge.

Our goal is to study the finite-time convergence of the error $\left\|Q_t^A - Q^*\right\|$ between one Q-estimator and the optimal Q-function (this is without the loss of generality due to the symmetry of the two estimators). To this end, our proof includes: (a) Part I which analyzes the stochastic error propagation between the two Q-estimators $\left\|Q_t^B - Q_t^A\right\|$; (b) Part II which analyzes the error dynamics between one Q-estimator and the optimum $\left\|Q_t^A - Q^*\right\|$ conditioned on the error event in Part I; and (c) Part III which bounds the unconditional error $\left\|Q_t^A - Q^*\right\|$. We describe each of the three parts in more details below.

**Part I: Bounding $\left\|Q_t^B - Q_t^A\right\|$ (see Proposition 1).** The main idea is to upper bound $\left\|Q_t^B - Q_t^A\right\|$ by a decreasing sequence $\{G_q\}_{q \geq 0}$ block-wisely with high probability, where each block $q$ (with $q \geq 0$) is defined by $t \in [\hat{\tau}_q, \hat{\tau}_{q+1})$. The proof consists of the following four steps.

*Step 1 (see Lemma 2)*: We characterize the dynamics of $u_t^{BA}(s,a) := Q^B(s,a) - Q^A(s,a)$ as an SA algorithm as follows:

$$u_{t+1}^{BA}(s,a) = (1-\alpha_t)u_t^{BA}(s,a) + \alpha_t(h_t(s,a) + z_t(s,a)),$$

where $h_t$ is a contractive mapping of $u_t^{BA}$, and $z_t$ is a martingale difference sequence.

*Step 2 (see Lemma 3)*: We derive lower and upper bounds on $u_t^{BA}$ via two sequences $X_{t;\hat{\tau}_q}$ and $Z_{t;\hat{\tau}_q}$ as follows:

$$-X_{t;\hat{\tau}_q}(s,a) + Z_{t;\hat{\tau}_q}(s,a) \leq u_t^{BA}(s,a) \leq X_{t;\hat{\tau}_q}(s,a) + Z_{t;\hat{\tau}_q}(s,a),$$

for any $t \geq \hat{\tau}_q$, state-action pair $(s,a) \in \mathcal{S} \times \mathcal{A}$, and $q \geq 0$, where $X_{t;\hat{\tau}_q}$ is deterministic and driven by $G_q$, and $Z_{t;\hat{\tau}_q}$ is stochastic and driven by the martingale difference sequence $z_t$.

*Step 3 (see Lemma 5 and Lemma 6)*: We block-wisely bound $u_t^{BA}(s,a)$ using the induction arguments. Namely, we prove $\left\|u_t^{BA}\right\| \leq G_q$ for $t \in [\hat{\tau}_q, \hat{\tau}_{q+1})$ holds for all $q \geq 0$. By induction, we first observe for $q = 0$, $\left\|u_t^{BA}\right\| \leq G_0$ holds. Given any state-action pair $(s,a)$, we assume that $\left\|u_t^{BA}(s,a)\right\| \leq G_q$ holds for $t \in [\hat{\tau}_q, \hat{\tau}_{q+1})$. Then we show $\left\|u_t^{BA}(s,a)\right\| \leq G_{q+1}$ holds for $t \in [\hat{\tau}_{q+1}, \hat{\tau}_{q+2})$, which follows by bounding $X_{t;\hat{\tau}_q}$ and $Z_{t;\hat{\tau}_q}$ separately in Lemma 5 and Lemma 6, respectively.

*Step 4 (see Appendix C.1.4)*: We apply union bound (Lemma 8) to obtain the block-wise bound for all state-action pairs and all blocks.

**Part II: Conditionally bounding $\left\|Q_t^A - Q^*\right\|$ (see Proposition 2).** We upper bound $\left\|Q_t^A - Q^*\right\|$ by a decreasing sequence $\{D_k\}_{k \geq 0}$ block-wisely conditioned on the following two events:

Event $E$: $\left\|u_t^{BA}\right\|$ is upper bounded properly (see (12) in Appendix C.2), and

Event $F$: there are sufficient updates of $Q_t^A$ in each block (see (13) in Appendix C.2).

The proof of Proposition 2 consists of the following four steps.

*Step 1 (see Lemma 10)*: We design a special relationship (illustrated in Figure 1) between the block-wise bounds $\{G_q\}_{q \geq 0}$ and $\{D_k\}_{k \geq 0}$ and their block separations.

*Step 2 (see Lemma 11)*: We characterize the dynamics of the iteration residual $r_t(s,a) := Q_t^A(s,a) - Q^*(s,a)$ as an SA algorithm as follows: when $Q^A$ is chosen to be updated at iteration $t$,

$$r_{t+1}(s,a) = (1-\alpha_t)r_t(s,a) + \alpha_t(\mathcal{T}Q_t^A(s,a) - Q^*(s,a)) + \alpha_t w_t(s,a) + \alpha_t \gamma u_t^{BA}(s',a^*),$$

where $w_t(s,a)$ is the error between the Bellman operator and the sample-based empirical estimator, and is thus a martingale difference sequence, and $u_t^{BA}$ has been defined in Part I.

*Step 3 (see Lemma 12)*: We provide upper and lower bounds on $r_t$ via two sequences $Y_{t;\tau_k}$ and $W_{t;\tau_k}$ as follows:

$$-Y_{t;\tau_k}(s,a) + W_{t;\tau_k}(s,a) \leq r_t(s,a) \leq Y_{t;\tau_k}(s,a) + W_{t;\tau_k}(s,a),$$

for all $t \geq \tau_k$, all state-action pairs $(s,a) \in \mathcal{S} \times \mathcal{A}$, and all $q \geq 0$, where $Y_{t;\tau_k}$ is deterministic and driven by $D_k$, and $W_{t;\tau_k}$ is stochastic and driven by the martingale difference sequence $w_t$. In particular, if $Q_t^A$ is not updated at some iteration, then the sequences $Y_{t;\tau_k}$ and $W_{t;\tau_k}$ assume the same values from the previous iteration.

*Step 4 (see Lemma 13, Lemma 14 and Appendix C.2.4)*: Similarly to Steps 3 and 4 in Part I, we conditionally bound $\|r_t\| \leq D_k$ for $t \in [\tau_k, \tau_{k+1})$ and $k \geq 0$ via bounding $Y_{t;\tau_k}$ and $W_{t;\tau_k}$ and further taking the union bound.

**Part III: Bounding $\left\|Q_t^A - Q^*\right\|$ (see Appendix C.3).** We combine the results in the first two parts, and provide high probability bound on $\|r_t\|$ with further probabilistic arguments, which exploit the high probability bounds on $\mathbb{P}(E)$ in Proposition 1 and $\mathbb{P}(F)$ in Lemma 15.

## 4 Conclusion

In this paper, we provide the first finite-time results for double Q-learning, which characterize how fast double Q-learning converges under both synchronous and asynchronous implementations. For the synchronous case, we show that it achieves an $\epsilon$-accurate optimal Q-function with at least the probability of $1 - \delta$ by taking $\Omega\left(\left(\frac{1}{(1-\gamma)^6\epsilon^2}\ln\frac{|\mathcal{S}||\mathcal{A}|}{(1-\gamma)^7\epsilon^2\delta}\right)^{\frac{1}{\omega}} + \left(\frac{1}{1-\gamma}\ln\frac{1}{(1-\gamma)^2\epsilon}\right)^{\frac{1}{1-\omega}}\right)$ iterations. Similar scaling order on $\frac{1}{1-\gamma}$ and $\frac{1}{\epsilon}$ also applies for asynchronous double Q-learning but with extra dependence on the covering number. We develop new techniques to bound the error between two correlated stochastic processes, which can be of independent interest.

## Broader Impact

Reinforcement learning has achieved great success in areas such as robotics and game playing, and thus has aroused broad interests and more potential real-world applications. Double Q-learning is a commonly used technique in deep reinforcement learning to improve the implementation stability and speed of deep Q-learning. In this paper, we provided the fundamental analysis on the convergence rate for double Q-learning, which theoretically justified the empirical success of double Q-learning in practice. Such a theory also provides practitioners desirable performance guarantee to further develop such a technique into various transferable technologies.

## Acknowledgements

The work was supported in part by the U.S. National Science Foundation under the grant CCF-1761506, National University of Singapore startup grant R-263-000-E60-133, National Natural Science Foundation of China (Grant No. 62073159), and the Shenzhen Science and Technology Program (Grant No. JCYJ20200109141601708).

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
