[Supplementary Material]

# Supplementary Materials

## A   Numerical Example on Convergence Bounds

We use the following numerical experiment to further illustrate our finite-time bounds on the convergence of double Q-learning. Specifically, we execute the synchronous double Q-learning algorithm over the MDP model of Grid World used in the experiment in Wainwright (2019). We adopt a random reward function which has a uniformly distribution over $\{R_{sa} - 20, R_{sa} + 20\}$, where $R_{sa} = 1$ is the expected reward. We run the same experiment 20 times independently with each experiment taking $3 \cdot 10^6$ iterations. In addition, we initialize both Q estimators as $Q^A = Q^B = -80 \cdot [1]^{|\mathcal{S}| \times |\mathcal{A}|}$, where $[1]^{|\mathcal{S}| \times |\mathcal{A}|}$ denotes the all-one matrix with the dimension of $|\mathcal{S}| \times |\mathcal{A}|$.

In such an experiment, the optimal Q-function can be explicitly calculated and thus the learning errors can be tracked. Figure 2 shows how the two errors $\left\|Q_t^A - Q^*\right\|$ and $\left\|Q_t^A - Q_t^B\right\|$ decay as the number of iterations enlarges experimentally. To further illustrate our theoretical bounds, we apply $\tau_1 = 800$ and $\tau_{k+1} = \tau_k + 500\tau_k^{0.5}$ to compute the block-wise upper bounds, and plot them also in Figure 2. It can be seen that the actual experimental errors are bounded by their corresponding block-wise upper bounds $\{D_k\}$ and $\{G_q\}$.

Figure 2: The convergence errors and their theoretical bounds of double Q-learning in the Grid World experiment. We choose $\gamma = 0.8, \alpha_t = \frac{1}{t^{0.85}}$ and $\tau_{k+1} = \tau_k + 500\tau_k^{0.5}$ with $\tau_1 = 800$.

## B   Proof of Lemma 1

We prove Lemma 1 by induction. First, it is easy to justify that the initial case is satisfied, i.e., $\left\|Q_1^A\right\| \le \frac{R_{\max}}{1-\gamma} = \frac{V_{\max}}{2}, \left\|Q_1^B\right\| \le \frac{V_{\max}}{2}$. (In practice we usually initialize the algorithm as $Q_1^A = Q_1^B = 0$). Next, we assume that at time t $\left\|Q_t^A\right\| \le \frac{V_{\max}}{2}$ and $\left\|Q_t^B\right\| \le \frac{V_{\max}}{2}$ hold. It then remains to show that these conditions also hold for $t + 1$. To this end, we observe that

$$
\begin{aligned}
\left\|Q_{t+1}^A(s,a)\right\| &= \left\|(1-\alpha_t)Q_t^A(s,a) + \alpha_t \left(R_t + \gamma Q_t^B\left(s', \arg\max_{a' \in U(s')} Q_t^A(s',a')\right)\right)\right\| \\
&\le (1-\alpha_t)\left\|Q_t^A\right\| + \alpha_t\left\|R_t\right\| + \alpha_t\gamma\left\|Q_t^B\right\| \\
&\le (1-\alpha_t)\frac{R_{\max}}{1-\gamma} + \alpha_t R_{\max} + \frac{\alpha_t\gamma R_{\max}}{1-\gamma} \\
&= \frac{R_{\max}}{1-\gamma} = \frac{V_{\max}}{2}.
\end{aligned}
$$

Similarly, we can show that $\left\|Q_{t+1}^B(s,a)\right\| \le \frac{V_{\max}}{2}$, which completes the proof.

# C  Proof of Theorem 1

In this appendix, we will provide a detailed proof of Theorem 1. Our proof includes: (a) Part I which analyzes the stochastic error propagation between the two Q-estimators $\left\|Q_t^B - Q_t^A\right\|$; (b) Part II which analyzes the error dynamics between one Q-estimator and the optimum $\left\|Q_t^A - Q^*\right\|$ conditioned on the error event in Part I; and (c) Part III which bounds the unconditional error $\left\|Q_t^A - Q^*\right\|$. We describe each of the three parts in more details below.

## C.1  Part I: Bounding $\left\|Q_t^B - Q_t^A\right\|$

The main idea is to upper bound $\left\|Q_t^B - Q_t^A\right\|$ by a decreasing sequence $\{G_q\}_{q \geq 0}$ block-wisely with high probability, where each block or epoch $q$ (with $q \geq 0$) is defined by $t \in [\hat{\tau}_q, \hat{\tau}_{q+1})$.

**Proposition 1.** *Fix $\epsilon > 0, \kappa \in (0,1), \sigma \in (0,1)$ and $\Delta \in (0, e-2)$. Consider synchronous double Q-learning using a polynomial learning rate $\alpha_t = \frac{1}{t^\omega}$ with $\omega \in (0,1)$. Let $G_q = (1-\xi)^q G_0$ with $G_0 = V_{\max}$ and $\xi = \frac{1-\gamma}{4}$. Let $\hat{\tau}_{q+1} = \hat{\tau}_q + \frac{2c}{\kappa}\hat{\tau}_q^\omega$ for $q \geq 1$ with $c \geq \frac{\ln(2+\Delta)+1/\hat{\tau}_1^\omega}{1-\ln(2+\Delta)-1/\hat{\tau}_1^\omega}$ and $\hat{\tau}_1$ as the finishing time of the first epoch satisfying*

$$\hat{\tau}_1 \geq \max\left\{ \left(\frac{1}{1-\ln(2+\Delta)}\right)^{\frac{1}{\omega}}, \left(\frac{128c(c+\kappa)V_{\max}^2}{\kappa^2\left(\frac{\Delta}{2+\Delta}\right)^2 \sigma^2\xi^2\epsilon^2} \ln\left(\frac{64c(c+\kappa)V_{\max}^2}{\kappa^2\left(\frac{\Delta}{2+\Delta}\right)^2 \sigma^2\xi^2\epsilon^2}\right)\right)^{\frac{1}{\omega}} \right\}.$$

*Then for any $n$ such that $G_n \geq \sigma\epsilon$, we have*

$$\mathbb{P}\left[\forall q \in [0,n], \forall t \in [\hat{\tau}_{q+1}, \hat{\tau}_{q+2}), \left\|Q_t^B - Q_t^A\right\| \leq G_{q+1}\right]$$

$$\geq 1 - \frac{4c(n+1)}{\kappa}\left(1 + \frac{2c}{\kappa}\right)|\mathcal{S}||\mathcal{A}|\exp\left(-\frac{\kappa^2\left(\frac{\Delta}{2+\Delta}\right)^2 \xi^2\sigma^2\epsilon^2\hat{\tau}_1^\omega}{64c(c+\kappa)V_{\max}^2}\right).$$

The proof of Proposition 1 consists of the following four steps.

### C.1.1  Step 1: Characterizing the dynamics of $Q_t^B(s,a) - Q_t^A(s,a)$

We first characterize the dynamics of $u_t^{BA}(s,a) := Q_t^B(s,a) - Q_t^A(s,a)$ as a stochastic approximation (SA) algorithm in this step.

**Lemma 2.** *Consider double Q-learning in Algorithm 1. Then we have*

$$u_{t+1}^{BA}(s,a) = (1-\alpha_t)u_t^{BA}(s,a) + \alpha_t F_t(s,a),$$

*where*

$$F_t(s,a) = \begin{cases} Q_t^B(s,a) - R_t - \gamma Q_t^B(s_{t+1}, a^*), & \text{w.p. 1/2} \\ R_t + \gamma Q_t^A(s_{t+1}, b^*) - Q_t^A(s,a), & \text{w.p. 1/2}. \end{cases}$$

*In addition, $F_t$ satisfies*

$$\|\mathbb{E}[F_t|\mathcal{F}_t]\| \leq \frac{1+\gamma}{2}\left\|u_t^{BA}\right\|,$$

*where the filtration $\mathcal{F}$ in the synchronous case is given by $\mathcal{F}_t = \sigma\left(\{s_k\}, \{R_{k-1}\}, 2 \leq k \leq t\right)$.*

*Proof.* Algorithm 1 indicates that at each time, either $Q^A$ or $Q^B$ is updated with equal probability. When updating $Q^A$ at time $t$, for each $(s,a)$ we have

$$\begin{aligned} u_{t+1}^{BA}(s,a) &= Q_{t+1}^B(s,a) - Q_{t+1}^A(s,a) \\ &= Q_t^B(s,a) - (Q_t^A(s,a) + \alpha_t(R_t + \gamma Q_t^B(s_{t+1}, a^*) - Q_t^A(s,a))) \\ &= (1-\alpha_t)Q_t^B(s,a) - ((1-\alpha_t)Q_t^A(s,a) + \alpha_t(R_t + \gamma Q_t^B(s_{t+1}, a^*) - Q_t^B(s,a))) \\ &= (1-\alpha_t)u_t^{BA}(s,a) + \alpha_t(Q_t^B(s,a) - R_t - \gamma Q_t^B(s_{t+1}, a^*)). \end{aligned}$$

Similarly, when updating $Q^B$, we have

$$u_{t+1}^{BA}(s,a) = Q_{t+1}^B(s,a) - Q_{t+1}^A(s,a)$$
$$= (Q_t^B(s,a) + \alpha_t(R_t + \gamma Q_t^A(s_{t+1}, b^*) - Q_t^B(s,a))) - Q_t^A(s,a)$$
$$= (1 - \alpha_t)Q_t^B(s,a) + (\alpha_t(R_t + \gamma Q_t^A(s_{t+1}, b^*) - Q_t^A(s,a)) - (1 - \alpha_t)Q_t^A(s,a))$$
$$= (1 - \alpha_t)u_t^{BA}(s,a) + \alpha_t(R_t + \gamma Q_t^A(s_{t+1}, b^*) - Q_t^A(s,a)).$$

Therefore, we can rewrite the dynamics of $u_t^{BA}$ as $u_{t+1}^{BA}(s,a) = (1 - \alpha_t)u_t^{BA}(s,a) + \alpha_t F_t(s,a)$, where

$$F_t(s,a) = \begin{cases} Q_t^B(s,a) - R_t - \gamma Q_t^B(s_{t+1}, a^*), & \text{w.p. } 1/2 \\ R_t + \gamma Q_t^A(s_{t+1}, b^*) - Q_t^A(s,a), & \text{w.p. } 1/2. \end{cases}$$

Thus, we have

$$\mathbb{E}[F_t(s,a)|\mathcal{F}_t]$$
$$= \frac{1}{2}\left(Q_t^B(sa) - \mathbb{E}_{s_{t+1}}[R_{sa}^{s'} - \gamma Q_t^B(s_{t+1}, a^*)]\right) + \frac{1}{2}\left(\mathbb{E}_{s_{t+1}}[R_{s,a}^{s'} + \gamma Q_t^A(s_{t+1}, b^*)] - Q_t^A(s,a)\right)$$
$$= \frac{1}{2}(Q_t^B(s,a) - Q_t^A(s,a)) + \frac{\gamma}{2}\mathbb{E}_{s_{t+1}}\left[Q_t^A(s_{t+1}, b^*) - Q_t^B(s_{t+1}, a^*)\right]$$
$$= \frac{1}{2}u_t^{BA}(s,a) + \frac{\gamma}{2}\mathbb{E}_{s_{t+1}}\left[Q_t^A(s_{t+1}, b^*) - Q_t^B(s_{t+1}, a^*)\right]. \tag{7}$$

Next, we bound $\mathbb{E}_{s_{t+1}}\left[Q_t^A(s_{t+1}, b^*) - Q_t^B(s_{t+1}, a^*)\right]$. First, consider the case when $\mathbb{E}_{s_{t+1}} Q_t^A(s_{t+1}, b^*) \geq \mathbb{E}_{s_{t+1}} Q_t^B(s_{t+1}, a^*)$. Then we have

$$\left|\mathbb{E}_{s_{t+1}}\left[Q_t^A(s_{t+1}, b^*) - Q_t^B(s_{t+1}, a^*)\right]\right| = \mathbb{E}_{s_{t+1}}\left[Q_t^A(s_{t+1}, b^*) - Q_t^B(s_{t+1}, a^*)\right]$$
$$\overset{(i)}{\leq} \mathbb{E}_{s_{t+1}}\left[Q_t^A(s_{t+1}, a^*) - Q_t^B(s_{t+1}, a^*)\right]$$
$$\leq \|u_t^{BA}\|,$$

where (i) follow from the definition of $a^*$ in Algorithm 1. Similarly, if $\mathbb{E}_{s_{t+1}} Q_t^A(s_{t+1}, b^*) < \mathbb{E}_{s_{t+1}} Q_t^B(s_{t+1}, a^*)$, we have

$$\left|\mathbb{E}_{s_{t+1}}\left[Q_t^A(s_{t+1}, b^*) - Q_t^B(s_{t+1}, a^*)\right]\right| = \mathbb{E}_{s_{t+1}}\left[Q_t^B(s_{t+1}, a^*) - Q_t^A(s_{t+1}, b^*)\right]$$
$$\overset{(i)}{\leq} \mathbb{E}_{s_{t+1}}\left[Q_t^B(s_{t+1}, b^*) - Q_t^A(s_{t+1}, b^*)\right]$$
$$\leq \|u_t^{BA}\|,$$

where (i) follows from the definition of $b^*$. Thus we can conclude that

$$\left|\mathbb{E}_{s_{t+1}}\left[Q_t^A(s_{t+1}, b^*) - Q_t^B(s_{t+1}, a^*)\right]\right| \leq \|u_t^{BA}\|.$$

Then, we continue to bound (7), and obtain

$$|\mathbb{E}[F_t(s,a)|\mathcal{F}_t]| = \left|\frac{1}{2}u_t^{BA}(s,a) + \frac{\gamma}{2}\mathbb{E}_{s_{t+1}}\left[Q_t^A(s_{t+1}, b^*) - Q_t^B(s_{t+1}, a^*)\right]\right|$$
$$\leq \frac{1}{2}\|u_t^{BA}\| + \frac{\gamma}{2}\left|\mathbb{E}_{s_{t+1}}\left[Q_t^A(s_{t+1}, b^*) - Q_t^B(s_{t+1}, a^*)\right]\right|$$
$$\leq \frac{1+\gamma}{2}\|u_t^{BA}\|,$$

for all $(s,a)$ pairs. Hence, $\|\mathbb{E}[F_t|\mathcal{F}_t]\| \leq \frac{1+\gamma}{2}\|u_t^{BA}\|$. $\qquad\square$

Applying Lemma 2, we write the dynamics of $u_t^{BA}(s,a)$ in the form of a classical SA algorithm driven by a martingale difference sequence as follows:

$$u_{t+1}^{BA}(s,a) = (1-\alpha_t)u_t^{BA}(s,a) + \alpha_t F_t(s,a) = (1-\alpha_t)u_t^{BA}(s,a) + \alpha_t(h_t(s,a) + z_t(s,a)),$$

where $h_t(s,a) = \mathbb{E}[F_t(s,a)|\mathcal{F}_t]$ and $z_t(s,a) = F_t(s,a) - \mathbb{E}[F_t|\mathcal{F}_t]$. Then, we obtain $\mathbb{E}[z_t(s,a)|\mathcal{F}_t] = 0$ and $\|h_t\| \le \frac{1+\gamma}{2}\|u_t^{BA}\|$ following from Lemma 2. We define $u^*(s,a) = 0$, and treat $h_t$ as an operator over $u_t^{BA}$. Then $h_t$ has a contraction property as:

$$\|h_t - u^*\| \le \gamma' \|u_t^{BA} - u^*\|, \tag{8}$$

where $\gamma' = \frac{1+\gamma}{2} \in (0,1)$. Based on this SA formulation, we bound $u_t^{BA}(s,a)$ block-wisely in the next step.

### C.1.2 Step 2: Constructing sandwich bounds on $u_t^{BA}$

We derive lower and upper bounds on $u_t^{BA}$ via two sequences $X_{t;\hat{\tau}_q}$ and $Z_{t;\hat{\tau}_q}$ in the following lemma.

**Lemma 3.** *Let $\hat{\tau}_q$ be such that $\|u_t^{BA}\| \le G_q$ for all $t \ge \hat{\tau}_q$. Define $Z_{t;\hat{\tau}_q}(s,a), X_{t;\hat{\tau}_q}(s,a)$ as*

$$Z_{t+1;\hat{\tau}_q}(s,a) = (1-\alpha_t)Z_{t;\hat{\tau}_q}(s,a) + \alpha_t z_t(s,a), \quad \text{with } Z_{\hat{\tau}_q;\hat{\tau}_q}(s,a) = 0;$$

$$X_{t+1;\hat{\tau}_q}(s,a) = (1-\alpha_t)X_{t;\hat{\tau}_q}(s,a) + \alpha_t \gamma' G_q, \quad \text{with } X_{\hat{\tau}_q;\hat{\tau}_q}(s,a) = G_q, \gamma' = \frac{1+\gamma}{2}.$$

*Then for any $t \ge \hat{\tau}_q$ and state-action pair $(s,a)$, we have*

$$-X_{t;\hat{\tau}_q}(s,a) + Z_{t;\hat{\tau}_q}(s,a) \le u_t^{BA}(s,a) \le X_{t;\hat{\tau}_q}(s,a) + Z_{t;\hat{\tau}_q}(s,a).$$

*Proof.* We proceed the proof by induction. For the initial condition $t = \hat{\tau}_q$, $\|u_{\hat{\tau}_q}^{BA}\| \le G_q$ implies $-G_q \le u_{\hat{\tau}_q}^{BA} \le G_q$. We assume the sandwich bound holds for time $t$. It remains to check that the bound also holds for $t+1$.

At time $t+1$, we have

$$
\begin{aligned}
u_{t+1}^{BA}(s,a) &= (1-\alpha_t)u_t^{BA}(s,a) + \alpha_t(h_t(s,a) + z_t(s,a)) \\
&\le (1-\alpha_t)(X_{t;\hat{\tau}_q}(s,a) + Z_{t;\hat{\tau}_q}(s,a)) + \alpha_t(h_t(s,a) + z_t(s,a)) \\
&\overset{(i)}{\le} \left[(1-\alpha_t)X_{t;\hat{\tau}_q}(s,a) + \alpha_t \gamma'\|u_t^{BA}\|\right] + \left[(1-\alpha_t)Z_{t;\hat{\tau}_q}(s,a) + \alpha_t z_t(s,a)\right] \\
&\le \left[(1-\alpha_t)X_{t;\hat{\tau}_q}(s,a) + \alpha_t \gamma' G_q\right] + \left[(1-\alpha_t)Z_{t;\hat{\tau}_q}(s,a) + \alpha_t z_t(s,a)\right] \\
&= X_{t+1;\hat{\tau}_q}(s,a) + Z_{t+1;\hat{\tau}_q}(s,a),
\end{aligned}
$$

where (i) follows from Lemma 2. Similarly, we can bound the other direction as

$$
\begin{aligned}
u_{t+1}^{BA}(s,a) &= (1-\alpha_t)u_t^{BA}(s,a) + \alpha_t(h_t(s,a) + z_t(s,a)) \\
&\ge (1-\alpha_t)(-X_{t;\hat{\tau}_q}(s,a) + Z_{t;\hat{\tau}_q}(s,a)) + \alpha_t(h_t(s,a) + z_t(s,a)) \\
&\ge \left[-(1-\alpha_t)X_{t;\hat{\tau}_q}(s,a) - \alpha_t \gamma'\|u_t^{BA}\|\right] + \left[(1-\alpha_t)Z_{t;\hat{\tau}_q}(s,a) + \alpha_t z_t(s,a)\right] \\
&\ge \left[-(1-\alpha_t)X_{t;\hat{\tau}_q}(s,a) - \alpha_t \gamma' G_q\right] + \left[(1-\alpha_t)Z_{t;\hat{\tau}_q}(s,a) + \alpha_t z_t(s,a)\right] \\
&= -X_{t+1;\hat{\tau}_q}(s,a) + Z_{t+1;\hat{\tau}_q}(s,a).
\end{aligned}
$$

$\square$

### C.1.3 Step 3: Bounding $X_{t;\hat{\tau}_q}$ and $Z_{t;\hat{\tau}_q}$ for block $q+1$

We bound $X_{t;\hat{\tau}_q}$ and $Z_{t;\hat{\tau}_q}$ in Lemma 5 and Lemma 6 below, respectively. Before that, we first introduce the following technical lemma which will be useful in the proof of Lemma 5.

**Lemma 4.** *Fix $\omega \in (0,1)$. Let $0 < t_1 < t_2$. Then we have*

$$\prod_{i=t_1}^{t_2}\left(1 - \frac{1}{i^\omega}\right) \le \exp\left(-\frac{t_2 - t_1}{t_2^\omega}\right).$$

*Proof.* Since $\ln(1-x) \leq -x$ for any $x \in (0,1)$, we have

$$\ln\left[\prod_{i=t_1}^{t_2}\left(1-\frac{1}{i^\omega}\right)\right] \leq -\sum_{i=t_1}^{t_2} i^{-\omega} \leq -\int_{t_1}^{t_2} t^{-\omega}dt = -\frac{t_2^{1-\omega}-t_1^{1-\omega}}{1-\omega}.$$

Thus, fix $\omega \in (0,1)$, let $0 < t_1 < t_2$, and then we have

$$\prod_{i=t_1}^{t_2}\left(1-\frac{1}{i^\omega}\right) \leq \exp\left(-\frac{t_2^{1-\omega}-t_1^{1-\omega}}{1-\omega}\right).$$

Define $f(t) := t^{1-\omega}$. Observe that $f(t)$ is an increasing concave function. Then we have

$$t_2^{1-\omega}-t_1^{1-\omega} \geq f'(t_2)(t_2-t_1) = (1-\omega)t_2^{-\omega}(t_2-t_1),$$

which immediately indicates the result. $\qquad\square$

We now derive a bound for $X_{t;\hat{\tau}_q}$.

**Lemma 5.** *Fix $\kappa \in (0,1)$ and $\Delta \in (0, e-2)$. Let $\{G_q\}$ be defined in Proposition 1. Consider synchronous double Q-learning using a polynomial learning rate $\alpha_t = \frac{1}{t^\omega}$ with $\omega \in (0,1)$. Suppose that $X_{t;\hat{\tau}_q}(s,a) \leq G_q$ for any $t \geq \hat{\tau}_q$. Then for any $t \in [\hat{\tau}_{q+1}, \hat{\tau}_{q+2})$, given $\hat{\tau}_{q+1} = \hat{\tau}_q + \frac{2c}{\kappa}\hat{\tau}_q^\omega$ with $\hat{\tau}_1 \geq \left(\frac{1}{1-\ln(2+\Delta)}\right)^{\frac{1}{\omega}}$ and $c \geq \frac{\ln(2+\Delta)+1/\hat{\tau}_1^\omega}{1-\ln(2+\Delta)-1/\hat{\tau}_1^\omega}$, we have*

$$X_{t;\hat{\tau}_q}(s,a) \leq \left(\gamma' + \frac{2}{2+\Delta}\xi\right)G_q.$$

*Proof.* Observe that $X_{\hat{\tau}_q;\hat{\tau}_q}(s,a) = G_q = \gamma'G_q + (1-\gamma')G_q := \gamma'G_q + \rho_{\hat{\tau}_q}$. We can rewrite the dynamics of $X_{t;\hat{\tau}_q}(s,a)$ as

$$X_{t+1;\hat{\tau}_q}(s,a) = (1-\alpha_t)X_{t;\hat{\tau}_q}(s,a) + \alpha_t\gamma'G_q = \gamma'G_q + (1-\alpha_t)\rho_t,$$

where $\rho_{t+1} = (1-\alpha_t)\rho_t$. By the definition of $\rho_t$, we obtain

$$\rho_t = (1-\alpha_{t-1})\rho_{t-1} = \cdots = (1-\gamma')G_q\prod_{i=\hat{\tau}_q}^{t-1}(1-\alpha_i)$$

$$= (1-\gamma')G_q\prod_{i=\hat{\tau}_q}^{t-1}\left(1-\frac{1}{i^\omega}\right) \overset{(i)}{\leq} (1-\gamma')G_q\prod_{i=\hat{\tau}_q}^{\hat{\tau}_{q+1}-1}\left(1-\frac{1}{i^\omega}\right)$$

$$\overset{(ii)}{\leq} (1-\gamma')G_q\exp\left(-\frac{\hat{\tau}_{q+1}-1-\hat{\tau}_q}{(\hat{\tau}_{q+1}-1)^\omega}\right) \leq (1-\gamma')G_q\exp\left(-\frac{\hat{\tau}_{q+1}-1-\hat{\tau}_q}{\hat{\tau}_{q+1}^\omega}\right)$$

$$= (1-\gamma')G_q\exp\left(-\frac{\frac{2c}{\kappa}\hat{\tau}_q^\omega - 1}{\hat{\tau}_{q+1}^\omega}\right) = (1-\gamma')G_q\exp\left(-\frac{2c}{\kappa}\left(\frac{\hat{\tau}_q}{\hat{\tau}_{q+1}}\right)^\omega + \frac{1}{\hat{\tau}_{q+1}^\omega}\right)$$

$$\overset{(iii)}{\leq} (1-\gamma')G_q\exp\left(-\frac{2c}{\kappa}\frac{1}{1+\frac{2c}{\kappa}} + \frac{1}{\hat{\tau}_1^\omega}\right) \overset{(iv)}{\leq} (1-\gamma')G_q\exp\left(-\frac{c}{1+c} + \frac{1}{\hat{\tau}_1^\omega}\right),$$

where (i) follows because $\alpha_i$ is decreasing and $t \geq \hat{\tau}_{q+1}$, (ii) follows from Lemma 4, (iii) follows because $\hat{\tau}_q \geq \hat{\tau}_1$ and

$$\left(\frac{\hat{\tau}_q}{\hat{\tau}_{q+1}}\right)^\omega \geq \frac{\hat{\tau}_q}{\hat{\tau}_{q+1}} = \frac{\hat{\tau}_q}{\hat{\tau}_q + \frac{2c}{\kappa}\hat{\tau}_q^\omega} \geq \frac{1}{1+\frac{2c}{\kappa}},$$

and (iv) follows because $\frac{2c}{\kappa} \geq c$. Next, observing the conditions that $\hat{\tau}_1^\omega \geq \frac{1}{1-\ln(2+\Delta)}$ and $c \geq \frac{1}{1-\ln(2+\Delta)-1/\hat{\tau}_1^\omega} - 1$, we have

$$\frac{c}{1+c} - \frac{1}{\hat{\tau}_1^\omega} \geq \ln(2+\Delta).$$

Thus we have $\rho_t \leq \frac{1-\gamma'}{2+\Delta}G_q$. Finally, We finish our proof by further observing that $1-\gamma' = 2\xi$. $\quad\square$

Since we have bounded $X_{t;\hat{\tau}_q}(s,a)$ by $\left(\gamma' + \frac{2}{2+\Delta}\xi\right) G_q$ for all $t \geq \hat{\tau}_{q+1}$, it remains to bound $Z_{t;\hat{\tau}_q}(s,a)$ by $\left(1 - \frac{2}{2+\Delta}\right) \xi G_q$ for block $q+1$, which will further yield $\|u_t^{BA}(s,a)\| \leq (\gamma'+\xi)G_q = (1-\xi)G_q = G_{q+1}$ for any $t \in [\hat{\tau}_{q+1}, \hat{\tau}_{q+2})$ as desired. Differently from $X_{t;\hat{\tau}_q}(s,a)$ which is a deterministic monotonic sequence, $Z_{t;\hat{\tau}_q}(s,a)$ is stochastic. We need to capture the probability for a bound on $Z_{t;\hat{\tau}_q}(s,a)$ to hold for block $q + 1$. To this end, we introduce a different sequence $\{Z_{t;\hat{\tau}_q}^l(s,a)\}$ given by

$$Z_{t;\hat{\tau}_q}^l(s,a) = \sum_{i=\hat{\tau}_q}^{\hat{\tau}_q+l} \alpha_i \prod_{j=i+1}^{t-1} (1-\alpha_j)z_i(s,a) := \sum_{i=\hat{\tau}_q}^{\hat{\tau}_q+l} \phi_i^{q,t-1} z_i(s,a), \tag{9}$$

where $\phi_i^{q,t-1} = \alpha_i \prod_{j=i+1}^{t-1}(1-\alpha_j)$. By the definition of $Z_{t;\hat{\tau}_q}(s,a)$, one can check that $Z_{t;\hat{\tau}_q}(s,a) = Z_{t;\hat{\tau}_q}^{t-1-\hat{\tau}_q}(s,a)$. Thus we have

$$Z_{t;\hat{\tau}_q}(s,a) = Z_{t;\hat{\tau}_q}(s,a) - Z_{\hat{\tau}_q;\hat{\tau}_q}(s,a) = \sum_{l=1}^{t-1-\hat{\tau}_q} (Z_{t;\hat{\tau}_q}^l(s,a) - Z_{t;\hat{\tau}_q}^{l-1}(s,a)) + Z_{t;\hat{\tau}_q}^0(s,a). \tag{10}$$

In the following lemma, we capture an important property of $Z_{t;\hat{\tau}_q}^l(s,a)$ defined in (9).

**Lemma 6.** *For any $t \in [\hat{\tau}_{q+1}, \hat{\tau}_{q+2})$ and $1 \leq l \leq t-1-\hat{\tau}_q$, $Z_{t;\hat{\tau}_q}^l(s,a)$ is a martingale sequence and satisfies*

$$|Z_{t;\hat{\tau}_q}^l(s,a) - Z_{t;\hat{\tau}_q}^{l-1}(s,a)| \leq \frac{2V_{\max}}{\hat{\tau}_q^\omega}. \tag{11}$$

*Proof.* To show the martingale property, we observe that

$$\mathbb{E}[Z_{t;\hat{\tau}_q}^l(s,a) - Z_{t;\hat{\tau}_q}^{l-1}(s,a)|\mathcal{F}_{\hat{\tau}_q+l-1}] = \mathbb{E}[\phi_{\hat{\tau}_q+l}^{q,t-1} z_{\hat{\tau}_q+l}(s,a)|\mathcal{F}_{\hat{\tau}_q+l-1}]$$
$$= \phi_{\hat{\tau}_q+l}^{q,t-1}\mathbb{E}[z_{\hat{\tau}_q+l}(s,a)|\mathcal{F}_{\hat{\tau}_q+l-1}] = 0,$$

where the last equation follows from the definition of $z_t(s,a)$.

In addition, based on the definition of $\phi_i^{q,t-1}$ in (9) which requires $i \geq \hat{\tau}_q$, we have

$$\phi_i^{q,t-1} = \alpha_i \prod_{j=i+1}^{t-1} (1-\alpha_j) \leq \alpha_i \leq \frac{1}{\hat{\tau}_q^\omega}.$$

Further, since $|F_t| \leq \frac{2R_{\max}}{1-\gamma} = V_{\max}$, we obtain $|z_t(s,a)| = |F_t - \mathbb{E}[F_t|\mathcal{F}_t]| \leq 2V_{\max}$. Thus

$$|Z_{t;\hat{\tau}_q}^l(s,a) - Z_{t;\hat{\tau}_q}^{l-1}(s,a)| = \phi_{\hat{\tau}_q+l}^{q,t-1}|z_{\hat{\tau}_q+l}(s,a)| \leq \frac{2V_{\max}}{\hat{\tau}_q^\omega}.$$

$\square$

Lemma 6 guarantees that $Z_{t;\hat{\tau}_q}^l(s,a)$ is a martingale sequence, which allows us to apply the following Azuma's inequality.

**Lemma 7.** *(Azuma, 1967) Let $X_0, X_1, \ldots, X_n$ be a martingale sequence such that for each $1 \leq k \leq n$,*

$$|X_k - X_{k-1}| \leq c_k,$$

*where the $c_k$ is a constant that may depend on $k$. Then for all $n \geq 1$ and any $\epsilon > 0$,*

$$\mathbb{P}[|X_n - X_0| > \epsilon] \leq 2 \exp\left(-\frac{\epsilon^2}{2\sum_{k=1}^n c_k^2}\right).$$

By Azuma's inequality and the relationship between $Z_{t;\hat{\tau}_q}(s,a)$ and $Z_{t;\hat{\tau}_q}^l(s,a)$ in (9), we obtain

$$\mathbb{P}\left[|Z_{t;\hat{\tau}_q}(s,a)| > \hat{\epsilon}|t \in [\hat{\tau}_{q+1}, \hat{\tau}_{q+2}]\right]$$

$$\leq 2\exp\left(-\frac{\hat{\epsilon}^2}{2\sum_{l=1}^{t-\hat{\tau}_q-1}\left(Z_{t;\hat{\tau}_q}^l(s,a) - Z_{t;\hat{\tau}_q}^{l-1}(s,a)\right)^2 + 2(Z_{t;\hat{\tau}_q}^0(s,a))^2}\right)$$

$$\overset{(i)}{\leq} 2\exp\left(-\frac{\hat{\epsilon}^2\hat{\tau}_q^{2\omega}}{8(t-\hat{\tau}_q)V_{\max}^2}\right) \leq 2\exp\left(-\frac{\hat{\epsilon}^2\hat{\tau}_q^{2\omega}}{8(\hat{\tau}_{q+2}-\hat{\tau}_q)V_{\max}^2}\right)$$

$$\overset{(ii)}{\leq} 2\exp\left(-\frac{\kappa^2\hat{\epsilon}^2\hat{\tau}_q^\omega}{32c(c+\kappa)V_{\max}^2}\right) = 2\exp\left(-\frac{\kappa^2\hat{\epsilon}^2\hat{\tau}_q^\omega}{32c(c+\kappa)V_{\max}^2}\right),$$

where (i) follows from Lemma 6, and (ii) follows because

$$\hat{\tau}_{q+2} - \hat{\tau}_q = \frac{2c}{\kappa}\hat{\tau}_{q+1}^\omega + \frac{2c}{\kappa}\hat{\tau}_q^\omega = \frac{2c}{\kappa}\left(\hat{\tau}_q + \frac{2c}{\kappa}\hat{\tau}_q^\omega\right)^\omega + \frac{2c}{\kappa}\hat{\tau}_q^\omega \leq \frac{2c}{\kappa}\left(2 + \frac{2c}{\kappa}\right)\hat{\tau}_q^\omega = \frac{4c(c+\kappa)}{\kappa^2}\hat{\tau}_q^\omega.$$

### C.1.4 Step 4: Unionizing all blocks and state-action pairs

Now we are ready to prove Proposition 1 by taking a union of probabilities over all blocks and state-action pairs. Before that, we introduce the following two preliminary lemmas, which will be used for multiple times in the sequel.

**Lemma 8.** *Let $\{X_i\}_{i \in \mathcal{I}}$ be a set of random variables. Fix $\epsilon > 0$. If for any $i \in \mathcal{I}$, we have $\mathbb{P}(X_i \leq \epsilon) \geq 1 - \delta$, then*

$$\mathbb{P}(\forall i \in \mathcal{I}, X_i \leq \epsilon) \geq 1 - |\mathcal{I}|\delta.$$

*Proof.* By union bound, we have

$$\mathbb{P}(\forall i \in \mathcal{I}, X_i \leq \epsilon) = 1 - \mathbb{P}\left(\bigcup_{i \in \mathcal{I}} X_i > \epsilon\right) \geq 1 - \sum_{i \in \mathcal{I}}\mathbb{P}(X_i > \epsilon) \geq 1 - |\mathcal{I}|\delta.$$

□

**Lemma 9.** *Fix positive constants $a, b$ satisfying $2ab\ln ab > 1$. If $\tau \geq 2ab\ln ab$, then*

$$\tau^b\exp\left(-\frac{2\tau}{a}\right) \leq \exp\left(-\frac{\tau}{a}\right).$$

*Proof.* Let $c = ab$. If $\tau \leq c^2$, we have

$$c\ln\tau \leq c\ln c^2 = 2c\ln c \leq \tau.$$

If $\tau \geq c^2$, we have

$$c\ln\tau \leq \sqrt{\tau}\ln\tau \leq \sqrt{\tau}\sqrt{\tau} = \tau,$$

where the last inequality follows from $\ln x^2 = 2\ln x \leq x$. Therefore, we obtain $c\ln\tau = ab\ln\tau \leq \tau$. Thus $\tau^b \leq \exp\left(\frac{\tau}{a}\right)$, which implies this lemma.  □

### Proof of Proposition 1
Based on the results obtained above, we are ready to prove Proposition 1. Applying Lemma 8, we

have

$$\mathbb{P}\left[\forall(s,a), \forall q \in [0,n], \forall t \in [\hat{\tau}_{q+1}, \hat{\tau}_{q+2}), |Z_{t;\hat{\tau}_q}(s,a)| \leq \frac{\Delta}{2+\Delta}\xi G_q\right]$$

$$\geq 1 - \sum_{q=0}^{n} |\mathcal{S}||\mathcal{A}|(\hat{\tau}_{q+2} - \hat{\tau}_{q+1}) \cdot \mathbb{P}\left[|Z_{t;\hat{\tau}_q}(s,a)| > \frac{\Delta}{2+\Delta}\xi G_q \Big| t \in [\hat{\tau}_{q+1}, \hat{\tau}_{q+2})\right]$$

$$\geq 1 - \sum_{q=0}^{n} |\mathcal{S}||\mathcal{A}|\frac{2c}{\kappa}\hat{\tau}_{q+1}^{\omega} \cdot 2\exp\left(-\frac{\kappa^2\left(\frac{\Delta}{2+\Delta}\right)^2 \xi^2 G_q^2 \hat{\tau}_q^{\omega}}{32c(c+\kappa)V_{\max}^2}\right)$$

$$\geq 1 - \sum_{q=0}^{n} |\mathcal{S}||\mathcal{A}|\frac{2c}{\kappa}\left(1 + \frac{2c}{\kappa}\right)\hat{\tau}_q^{\omega} \cdot 2\exp\left(-\frac{\kappa^2\left(\frac{\Delta}{2+\Delta}\right)^2 \xi^2 G_q^2 \hat{\tau}_q^{\omega}}{32c(c+\kappa)V_{\max}^2}\right)$$

$$\overset{(i)}{\geq} 1 - \sum_{q=0}^{n} |\mathcal{S}||\mathcal{A}|\frac{2c}{\kappa}\left(1 + \frac{2c}{\kappa}\right)\hat{\tau}_q^{\omega} \cdot 2\exp\left(-\frac{\kappa^2\left(\frac{\Delta}{2+\Delta}\right)^2 \xi^2 \sigma^2 \epsilon^2 \hat{\tau}_q^{\omega}}{32c(c+\kappa)V_{\max}^2}\right)$$

$$\overset{(ii)}{\geq} 1 - \frac{4c}{\kappa}\left(1 + \frac{2c}{\kappa}\right)\sum_{q=0}^{n} |\mathcal{S}||\mathcal{A}| \cdot \exp\left(-\frac{\kappa^2\left(\frac{\Delta}{2+\Delta}\right)^2 \xi^2 \sigma^2 \epsilon^2 \hat{\tau}_q^{\omega}}{64c(c+\kappa)V_{\max}^2}\right)$$

$$\overset{(iii)}{\geq} 1 - \frac{4c(n+1)}{\kappa}\left(1 + \frac{2c}{\kappa}\right)|\mathcal{S}||\mathcal{A}|\exp\left(-\frac{\kappa^2\left(\frac{\Delta}{2+\Delta}\right)^2 \xi^2 \sigma^2 \epsilon^2 \hat{\tau}_1^{\omega}}{64c(c+\kappa)V_{\max}^2}\right),$$

where (i) follows because $G_q \geq G_n \geq \sigma\epsilon$, (ii) follows from Lemma 9 by substituting that $a = \frac{64c(c+\kappa)V_{\max}^2}{\kappa^2\left(\frac{\Delta}{2+\Delta}\right)^2\sigma^2\xi^2\epsilon^2}, b = 1$ and observing

$$\hat{\tau}_q^{\omega} \geq \hat{\tau}_1^{\omega} \geq \frac{128c(c+\kappa)V_{\max}^2}{\kappa^2\left(\frac{\Delta}{2+\Delta}\right)^2\sigma^2\xi^2\epsilon^2}\ln\left(\frac{64c(c+\kappa)V_{\max}^2}{\kappa^2\left(\frac{\Delta}{2+\Delta}\right)^2\sigma^2\xi^2\epsilon^2}\right) = 2ab\ln ab,$$

and (iii) follows because $\hat{\tau}_q \geq \hat{\tau}_1$.

Finally, we complete the proof of Proposition 1 by observing that $X_{t;\hat{\tau}_q}$ is a deterministic sequence and thus

$$\mathbb{P}\left[\forall q \in [0,n], \forall t \in [\hat{\tau}_{q+1}, \hat{\tau}_{q+2}), \left\|Q_t^B - Q_t^A\right\| \leq G_{q+1}\right]$$

$$\geq \mathbb{P}\left[\forall(s,a), \forall q \in [0,n], \forall t \in [\hat{\tau}_{q+1}, \hat{\tau}_{q+2}), |Z_{t;\hat{\tau}_q}(s,a)| \leq \frac{\Delta}{2+\Delta}\xi G_q\right].$$

## C.2 Part II: Conditionally bounding $\left\|Q_t^A - Q^*\right\|$

In this part, we upper bound $\left\|Q_t^A - Q^*\right\|$ by a decreasing sequence $\{D_k\}_{k\geq 0}$ block-wisely conditioned on the following two events: fix a positive integer $m$, we define

$$E := \left\{\forall k \in [0,m], \forall t \in [\tau_{k+1}, \tau_{k+2}), \left\|Q_t^B - Q_t^A\right\| \leq \sigma D_{k+1}\right\}, \tag{12}$$

$$F := \{\forall k \in [1, m+1], I_k^A \geq c\tau_k^{\omega}\}, \tag{13}$$

where $I_k^A$ denotes the number of iterations updating $Q^A$ at epoch $k$, $\tau_{k+1}$ is the starting iteration index of the $(k+1)$th block, and $\omega$ is the decay parameter of the polynomial learning rate. Roughly, Event $E$ requires that the difference between the two Q-estimators are bounded appropriately, and Event $F$ requires that $Q^A$ is sufficiently updated in each block.

**Proposition 2.** *Fix $\epsilon > 0, \kappa \in (\ln 2, 1)$ and $\Delta \in (0, e^{\kappa} - 2)$. Consider synchronous double Q-learning under a polynomial learning rate $\alpha_t = \frac{1}{t^{\omega}}$ with $\omega \in (0,1)$. Let $\{G_q\}_{q\geq 0}, \{\hat{\tau}_q\}_{q\geq 0}$ be*

*defined in Proposition 1. Define $D_k = (1-\beta)^k \frac{V_{\max}}{\sigma}$ with $\beta = \frac{1-\gamma(1+\sigma)}{2}$ and $\sigma = \frac{1-\gamma}{2\gamma}$. Let $\tau_k = \hat{\tau}_k$ for $k \geq 0$. Suppose that $c \geq \frac{\kappa(\ln(2+\Delta)+1/\tau_1^\omega)}{2(\kappa - \ln(2+\Delta) - 1/\tau_1^\omega)}$ and $\tau_1$ as the finishing time of the first block satisfies*

$$\tau_1 \geq \max \left\{ \left( \frac{1}{\kappa - \ln(2+\Delta)} \right)^{\frac{1}{\omega}}, \left( \frac{32c(c+\kappa)V_{\max}^2}{\kappa^2 \left( \frac{\Delta}{2+\Delta} \right)^2 \beta^2 \epsilon^2} \ln \left( \frac{16c(c+\kappa)V_{\max}^2}{\kappa^2 \left( \frac{\Delta}{2+\Delta} \right)^2 \beta^2 \epsilon^2} \right) \right)^{\frac{1}{\omega}} \right\}.$$

*Then for any $m$ such that $D_m \geq \epsilon$, we have*

$$\mathbb{P}\left[ \forall k \in [0,m], \forall t \in [\tau_{k+1}, \tau_{k+2}), \left\| Q_t^A - Q^* \right\| \leq D_{k+1} | E, F \right]$$

$$\geq 1 - \frac{4c(m+1)}{\kappa} \left( 1 + \frac{2c}{\kappa} \right) |\mathcal{S}||\mathcal{A}| \exp \left( -\frac{\kappa^2 \left( \frac{\Delta}{2+\Delta} \right)^2 \beta^2 \epsilon^2 \tau_1^\omega}{16c(c+\kappa)V_{\max}^2} \right),$$

*where the events $E, F$ are defined in (12) and (13), respectively.*

The proof of Proposition 2 consists of the following four steps.

### C.2.1 Step 1: Designing $\{D_k\}_{k \geq 0}$

The following lemma establishes the relationship (illustrated in Figure 1) between the block-wise bounds $\{G_q\}_{q \geq 0}$ and $\{D_k\}_{k \geq 0}$ and their block separations, such that Event $E$ occurs with high probability as a result of Proposition 1.

**Lemma 10.** *Let $\{G_q\}$ be defined in Proposition 1, and let $D_k = (1-\beta)^k \frac{V_{\max}}{\sigma}$ with $\beta = \frac{1-\gamma(1+\sigma)}{2}$ and $\sigma = \frac{1-\gamma}{2\gamma}$. Then we have*

$$\mathbb{P}\left[ \forall q \in [0,m], \forall t \in [\hat{\tau}_{q+1}, \hat{\tau}_{q+2}), \left\| Q_t^B - Q_t^A \right\| \leq G_{q+1} \right]$$
$$\leq \mathbb{P}\left[ \forall k \in [0,m], \forall t \in [\tau_{k+1}, \tau_{k+2}), \left\| Q_t^B - Q_t^A \right\| \leq \sigma D_{k+1} \right],$$

*given that $\tau_k = \hat{\tau}_k$.*

*Proof.* Based on our choice of $\sigma$, we have

$$\beta = \frac{1 - \gamma(1+\sigma)}{2} = \frac{1 - \gamma \cdot \frac{1+\gamma}{2\gamma}}{2} = \frac{1-\gamma}{4} = \xi.$$

Therefore, the decay rate of $D_k$ is the same as that of $G_q$. Further considering $G_0 = \sigma D_0$, we can make the sequence $\{\sigma D_k\}$ as an upper bound of $\{G_q\}$ for any time as long as we set the same starting point and ending point for each epoch. □

In Lemma 10, we make $G_k = \sigma D_k$ at any block $k$ and $\xi = \beta = \frac{1-\gamma}{4}$ by careful design of $\sigma$. In fact, one can choose any value of $\sigma \in (0, (1-\gamma)/\gamma)$ and design a corresponding relationship between $\tau_k$ and $\hat{\tau}_k$ as long as the sequence $\{\sigma D_k\}$ can upper bound $\{G_q\}$ for any time. For simplicity of presentation, we keep the design in Lemma 10.

### C.2.2 Step 2: Characterizing the dynamics of $Q_t^A(s,a) - Q^*(s,a)$

We characterize the dynamics of the iteration residual $r_t(s,a) := Q_t^A(s,a) - Q^*(s,a)$ as an SA algorithm in Lemma 11 below. Since not all iterations contribute to the error propagation due to the random update between the two Q-estimators, we introduce the following notations to label the valid iterations.

**Definition 1.** *We define $T^A$ as the collection of iterations updating $Q^A$. In addition, we denote $T^A(t_1, t_2)$ as the set of iterations updating $Q^A$ between time $t_1$ and $t_2$. That is,*

$$T^A(t_1, t_2) = \left\{ t : t \in [t_1, t_2] \text{ and } t \in T^A \right\}.$$

*Correspondingly, the number of iterations updating $Q^A$ between time $t_1$ and $t_2$ is the cardinality of $T^A(t_1, t_2)$ which is denoted as $|T^A(t_1, t_2)|$.*

**Lemma 11.** *Consider double Q-learning in Algorithm 1. Then we have*

$$r_{t+1}(s,a) = \begin{cases} r_t(s,a), & t \notin T^A; \\ (1-\alpha_t)r_t(s,a) + \alpha_t(\mathcal{T}Q_t^A(s,a) - Q^*(s,a)) + \alpha_t w_t(s,a) + \alpha_t\gamma u_t^{BA}(s',a^*), & t \in T^A, \end{cases}$$

*where* $w_t(s,a) = \mathcal{T}_t Q_t^A(s,a) - \mathcal{T}Q_t^A(s,a), u_t^{BA}(s,a) = Q_t^B(s,a) - Q_t^A(s,a).$

*Proof.* Following from Algorithm 1 and for $t \in T^A$, we have

$$
\begin{aligned}
Q_{t+1}^A&(s,a) \\
&= Q_t^A(s,a) + \alpha_t(R_t + \gamma Q_t^B(s',a^*) - Q_t^A(s,a)) \\
&= (1-\alpha_t)Q_t^A(s,a) + \alpha_t\left(R_t + \gamma Q_t^A(s',a^*)\right) + \alpha_t\left(\gamma Q_t^B(s',a^*) - \gamma Q_t^A(s',a^*)\right) \\
&\overset{(i)}{=} (1-\alpha_t)Q_t^A(s,a) + \alpha_t\left(\mathcal{T}_t Q_t^A(s,a) + \gamma u_t^{BA}(s',a^*)\right) \\
&= (1-\alpha_t)Q_t^A(s,a) + \alpha_t\mathcal{T}Q_t^A(s,a) + \alpha_t(\mathcal{T}_t Q_t^A(s,a) - \mathcal{T}Q_t^A(s,a)) + \alpha_t\gamma u_t^{BA}(s',a^*) \\
&= (1-\alpha_t)Q_t^A(s,a) + \alpha_t\mathcal{T}Q_t^A(s,a) + \alpha_t w_t(s,a) + \alpha_t\gamma u_t^{BA}(s',a^*),
\end{aligned}
$$

where (i) follows because we denote $\mathcal{T}_t Q_t^A(s,a) = R_t + \gamma Q_t^A(s',a^*)$. By subtracting $Q^*$ from both sides, we complete the proof. $\square$

### C.2.3 Step 3: Constructing sandwich bounds on $r_t(s,a)$

We provide upper and lower bounds on $r_t$ by constructing two sequences $Y_{t;\tau_k}$ and $W_{t;\tau_k}$ in the following lemma.

**Lemma 12.** *Let* $\tau_k$ *be such that* $\|r_t\| \le D_k$ *for all* $t \ge \tau_k$. *Suppose that we have* $\left\|u_t^{BA}\right\| \le \sigma D_k$ *with* $\sigma = \frac{1-\gamma}{2\gamma}$ *for all* $t \ge \tau_k$. *Define* $W_{t;\tau_k}(s,a)$ *as*

$$W_{t+1;\tau_k}(s,a) = \begin{cases} W_{t;\tau_k}(s,a), & t \notin T^A; \\ (1-\alpha_t)W_{t;\tau_k}(s,a) + \alpha_t w_t(s,a), & t \in T^A, \end{cases}$$

*where* $W_{\tau_k;\tau_k}(s,a) = 0$ *and define* $Y_{t;\tau_k}(s,a)$ *as*

$$Y_{t+1;\tau_k}(s,a) = \begin{cases} Y_{t;\tau_k}(s,a), & t \notin T^A; \\ (1-\alpha_t)Y_{t;\tau_k}(s,a) + \alpha_t\gamma'' D_k, & t \in T^A, \end{cases}$$

*where* $Y_{\tau_k;\tau_k}(s,a) = D_k$ *and* $\gamma'' = \gamma(1+\sigma)$. *Then for any* $t \ge \tau_k$ *and state-action pair* $(s,a)$, *we have*

$$-Y_{t;\tau_k}(s,a) + W_{t;\tau_k}(s,a) \le r_t(s,a) \le Y_{t;\tau_k}(s,a) + W_{t;\tau_k}(s,a).$$

*Proof.* We proceed the proof by induction. For the initial condition $t = \tau_k$, we have $\|r_t(s,a)\| \le D_k$, and thus it holds that $-D_k \le r_{\tau_k}(s,a) \le D_k$. We assume the sandwich bound holds for time $t \ge \tau_k$. It remains to check whether this bound holds for $t+1$.

If $t \notin T^A$, then $r_{t+1}(s,a) = r_t(s,a), W_{t+1;\tau_k}(s,a) = W_{t;\tau_k}(s,a), Y_{t+1;\tau_k}(s,a) = Y_{t;\tau_k}(s,a)$. Thus the sandwich bound still holds.

If $t \in T^A$, we have

$$
\begin{aligned}
r_{t+1}(s,a) &= (1-\alpha_t)r_t(s,a) + \alpha_t(\mathcal{T}Q_t^A(s,a) - Q^*(s,a)) + \alpha_t w_t(s,a) + \alpha_t\gamma u_t^{BA}(s',a^*) \\
&\le (1-\alpha_t)(Y_{t;\tau_k}(s,a) + W_{t;\tau_k}(s,a)) + \alpha_t\left\|\mathcal{T}Q_t^A - Q^*\right\| \\
&\quad + \alpha_t w_t(s,a) + \alpha_t\gamma\left\|u_t^{BA}\right\| \\
&\overset{(i)}{\le} (1-\alpha_t)(Y_{t;\tau_k}(s,a) + W_{t;\tau_k}(s,a)) + \alpha_t\gamma\left\|r_t\right\| \\
&\quad + \alpha_t w_t(s,a) + \alpha_t\gamma\left\|u_t^{BA}\right\| \\
&\overset{(ii)}{\le} (1-\alpha_t)Y_{t;\tau_k}(s,a) + \alpha_t\gamma(1+\sigma)D_k + (1-\alpha_t)W_{t;\tau_k}(s,a) + \alpha_t w_t(s,a) \\
&\le Y_{t+1;\tau_k}(s,a) + W_{t+1;\tau_k}(s,a),
\end{aligned}
$$

where (i) follows from the contraction property of the Bellman operator, and (ii) follows from the condition $\left\|u_t^{BA}\right\| \leq \sigma D_k$.

Similarly, we can bound the other direction as

$$
\begin{aligned}
r_{t+1}(s,a) &= (1-\alpha_t)r_t(s,a) + \alpha_t(\mathcal{T}Q_t^A(s,a) - Q^*(s,a)) + \alpha_t w_t(s,a) + \alpha_t \gamma u_t^{BA}(s',a^*) \\
&\geq (1-\alpha_t)(-Y_{t;\tau_k}(s,a) + W_{t;\tau_k}(s,a)) - \alpha_t \left\|\mathcal{T}Q_t^A - Q^*\right\| \\
&\quad + \alpha_t w_t(s,a) - \alpha_t \gamma \left\|u_t^{BA}\right\| \\
&\geq (1-\alpha_t)(Y_{t;\tau_k}(s,a) + W_{t;\tau_k}(s,a)) - \alpha_t \gamma \left\|r_t\right\| \\
&\quad + \alpha_t w_t(s,a) - \alpha_t \gamma \left\|u_t^{BA}\right\| \\
&\geq -(1-\alpha_t)Y_{t;\tau_k}(s,a) - \alpha_t \gamma (1+\sigma)D_k + (1-\alpha_t)W_{t;\tau_k}(s,a) + \alpha_t w_t(s,a) \\
&\geq -Y_{t+1;\tau_k}(s,a) + W_{t+1;\tau_k}(s,a).
\end{aligned}
$$

$\square$

### C.2.4  Step 4: Bounding $Y_{t;\tau_k}(s,a)$ and $W_{t;\tau_k}(s,a)$ for epoch $k+1$

Similarly to Steps 3 and 4 in Part I, we conditionally bound $\|r_t\| \leq D_k$ for $t \in [\tau_k, \tau_{k+1})$ and $k = 0, 1, 2, \ldots$ by the induction arguments followed by the union bound. We first bound $Y_{t;\tau_k}(s,a)$ and $W_{t;\tau_k}(s,a)$ in Lemma 13 and Lemma 14, respectively.

**Lemma 13.** *Fix $\kappa \in (\ln 2, 1)$ and $\Delta \in (0, e^\kappa - 2)$. Let $\{D_k\}$ be defined in Lemma 10. Consider synchronous double Q-learning using a polynomial learning rate $\alpha_t = \frac{1}{t^\omega}$ with $\omega \in (0,1)$. Suppose that $Y_{t;\tau_k}(s,a) \leq D_k$ for any $t \geq \tau_k$. At block $k$, we assume that there are at least $c\tau_k^\omega$ iterations updating $Q^A$, i.e., $|T^A(\tau_k, \tau_{k+1})| \geq c\tau_k^\omega$. Then for any $t \in [\tau_{k+1}, \tau_{k+2})$, we have*

$$
Y_{t;\tau_k}(s,a) \leq \left(\gamma'' + \frac{2}{2+\Delta}\beta\right) D_k.
$$

*Proof.* Since we have defined $\tau_k = \hat{\tau}_k$ in Lemma 10, we have $\tau_{k+1} = \tau_k + \frac{2c}{\kappa}\tau_k^\omega$.

Observe that $Y_{\tau_k;\tau_k}(s,a) = D_k = \gamma'' D_k + (1-\gamma'')D_k := \gamma'' D_k + \rho_{\tau_k}$. We can rewrite the dynamics of $Y_{t;\tau_k}(s,a)$ as

$$
Y_{t+1;\tau_k}(s,a) = \begin{cases} Y_{t;\tau_k}(s,a), & t \notin T^A \\ (1-\alpha_t)Y_{t;\tau_k}(s,a) + \alpha_t \gamma'' D_k = \gamma'' D_k + (1-\alpha_t)\rho_t, & t \in T^A \end{cases}
$$

where $\rho_{t+1} = (1-\alpha_t)\rho_t$ for $t \in T^A$. By the definition of $\rho_t$, we obtain

$$
\begin{aligned}
\rho_t &= \rho_{\tau_k} \prod_{i \in T^A(\tau_k, t-1)} (1-\alpha_i) = (1-\gamma'')D_k \prod_{i \in T^A(\tau_k, t-1)} (1-\alpha_i) \\
&= (1-\gamma'')D_k \prod_{i \in T^A(\tau_k, t-1)} \left(1 - \frac{1}{i^\omega}\right) \overset{(i)}{\leq} (1-\gamma'')D_k \prod_{i \in T^A(\tau_k, \tau_{k+1}-1)} \left(1 - \frac{1}{i^\omega}\right) \quad (14) \\
&\overset{(ii)}{\leq} (1-\gamma'')D_k \prod_{i=\tau_{k+1}-c\tau_k^\omega}^{\tau_{k+1}-1} \left(1 - \frac{1}{i^\omega}\right) \overset{(iii)}{\leq} (1-\gamma'')D_k \exp\left(-\frac{c\tau_k^\omega - 1}{(\tau_{k+1}-1)^\omega}\right) \\
&\leq (1-\gamma'')D_k \exp\left(-\frac{c\tau_k^\omega - 1}{\tau_{k+1}^\omega}\right) = (1-\gamma'')D_k \exp\left(-c\left(\frac{\tau_k}{\tau_{k+1}}\right)^\omega + \frac{1}{\tau_{k+1}^\omega}\right) \\
&\overset{(iv)}{\leq} (1-\gamma'')D_k \exp\left(-\frac{c}{1+\frac{2c}{\kappa}} + \frac{1}{\tau_1^\omega}\right),
\end{aligned}
$$

where (i) follows because $\alpha_i < 1$ and $t \geq \tau_{k+1}$, (ii) follows because $|T^A(\tau_k, \tau_{k+1}-1)| \geq c\tau_k^\omega$ where $T^A(t_1, t_2)$ and $|T^A(t_1, t_2)|$ are defined in Definition 1, (iii) follows from Lemma 9, and (iv) holds because $\tau + k \geq \tau_1$ and

$$
\left(\frac{\tau_k}{\tau_{k+1}}\right)^\omega \geq \frac{\tau_k}{\tau_{k+1}} = \frac{\tau_k}{\tau_k + \frac{2c}{\kappa}\tau_k^\omega} \geq \frac{1}{1+\frac{2c}{\kappa}}.
$$

Next we check the value of the power $-\frac{c}{1+\frac{2c}{\kappa}} + \frac{1}{\tau_1^\omega}$. Since $\kappa \in (\ln 2, 1)$ and $\Delta \in (0, e^\kappa - 2)$, we have $\ln(2 + \Delta) \in (0, \kappa)$. Further, observing $\tau_1^\omega > \frac{1}{\kappa - \ln(2+\Delta)}$, we obtain $\ln(2 + \Delta) + \frac{1}{\tau_1^\omega} \in (0, \kappa)$. Last, since $c \geq \frac{\kappa}{2}\left(\frac{1}{1-\frac{\ln(2+\Delta)+1/\tau_1^\omega}{\kappa}} - 1\right) = \frac{\kappa(\ln(2+\Delta)+1/\tau_1^\omega)}{2(\kappa-\ln(2+\Delta)-1/\tau_1^\omega)}$, we have $-\frac{c}{1+\frac{2c}{\kappa}} + \frac{1}{\tau_1^\omega} \leq -\ln(2+\Delta)$.

Thus, we have $\rho_t \leq \frac{1-\gamma''}{2+\Delta} D_k$. Finally, we finish our proof by further observing that $1 - \gamma'' = 2\beta$.

$\square$

It remains to bound $|W_{t;\tau_k}(s,a)| \leq \left(1 - \frac{2}{2+\Delta}\right)\beta D_k$ for $t \in [\tau_{k+1}, \tau_{k+2})$. Combining the bounds of $Y_{t;\tau_k}$ and $W_{t;\tau_k}$ yields $(\gamma'' + \beta)D_k = (1-\beta)D_k = D_{k+1}$. Since $W_{t;\tau_k}$ is stochastic, we need to derive the probability for the bound to hold. To this end, we first rewrite the dynamics of $W_{t;\tau_k}$ defined in Lemma 12 as

$$W_{t;\tau_k}(s,a) = \sum_{i \in T^A(\tau_k, t-1)} \alpha_i \prod_{j \in T^A(i+1, t-1)} (1 - \alpha_j) w_i(s,a).$$

Next, we introduce a new sequence $\{W_{t;\tau_k}^l(s,a)\}$ as

$$W_{t;\tau_k}^l(s,a) = \sum_{i \in T^A(\tau_k, \tau_k+l)} \alpha_i \prod_{j \in T^A(i+1, t-1)} (1 - \alpha_j) w_i(s,a).$$

Thus we have $W_{t;\tau_k}(s,a) = W_{t;\tau_k}^{t-1-\tau_k}(s,a)$. Then we have the following lemma.

**Lemma 14.** *For any $t \in [\tau_{k+1}, \tau_{k+2}]$ and $1 \leq l \leq t - \tau_k - 1$, $\{W_{t;\tau_k}^l(s,a)\}$ is a martingale sequence and satisfies*

$$|W_{t;\tau_k}^l(s,a) - W_{t;\tau_k}^{l-1}(s,a)| \leq \frac{V_{\max}}{\tau_k^\omega}.$$

*Proof.* Observe that

$$W_{t;\tau_k}^l(s,a) - W_{t;\tau_k}^{l-1}(s,a) = \begin{cases} 0, & \tau_k + l - 1 \notin T^A; \\ \alpha_{\tau_k+l} \prod_{j \in T^A(\tau_k+l+1, t-1)} (1 - \alpha_j) w_{\tau_k+l}(s,a), & \tau_k + l - 1 \in T^A. \end{cases}$$

Since $\mathbb{E}[w_t | \mathcal{F}_{t-1}] = 0$, we have

$$\mathbb{E}\left[W_{t;\tau_k}^l(s,a) - W_{t;\tau_k}^{l-1}(s,a) | \mathcal{F}_{\tau_k+l-1}\right] = 0.$$

Thus $\{W_{t;\tau_k}^l(s,a)\}$ is a martingale sequence. In addition, since $l \geq 1$ and $\alpha_t \in (0,1)$, we have

$$\alpha_{\tau_k+l} \prod_{j \in T^A(\tau_k+l+1, t-1)} (1 - \alpha_j) \leq \alpha_{\tau_k+l} \leq \alpha_{\tau_k} = \frac{1}{\tau_k^\omega}.$$

Further, we obtain $|w_t(s,a)| = |\mathcal{T}_t Q_t^A(s,a) - \mathcal{T}Q_t^A(s,a)| \leq \frac{2Q_{\max}}{1-\gamma} = V_{\max}$. Thus

$$|W_{t;\tau_k}^l(s,a) - W_{t;\tau_k}^{l-1}(s,a)| \leq \alpha_{\tau_k+l}|w_{\tau_k+l}(s,a)| \leq \frac{V_{\max}}{\tau_k^\omega}.$$

$\square$

Next, we bound $W_{t;\tau_k}(s,a)$. Fix $\tilde{\epsilon} > 0$. Then for any $t \in [\tau_{k+1}, \tau_{k+2})$, we have
$\mathbb{P}\left[|W_{t;\tau_k}(s,a)| > \tilde{\epsilon} | t \in [\tau_{k+1}, \tau_{k+2}), E, F\right]$

$$\overset{(i)}{\leq} 2\exp\left(\frac{-\tilde{\epsilon}^2}{2\sum_{l:\tau_k+l-1\in T^A(\tau_k, t-1)} \left(W_{t;\tau_k}^l(s,a) - W_{t;\tau_k}^{l-1}(s,a)\right)^2 + 2(W_{t;\tau_k}^{\min(T^A(\tau_k, t-1))}(s,a))^2}\right)$$

$$\overset{(ii)}{\leq} 2\exp\left(-\frac{\tilde{\epsilon}^2\tau_k^{2\omega}}{2(|T^A(\tau_k, t-1)|+1)V_{\max}^2}\right) \overset{(iii)}{\leq} 2\exp\left(-\frac{\tilde{\epsilon}^2\tau_k^{2\omega}}{2(t+1-\tau_k)V_{\max}^2}\right)$$

$$\leq 2\exp\left(-\frac{\tilde{\epsilon}^2\tau_k^{2\omega}}{2(\tau_{k+2}-\tau_k)V_{\max}^2}\right) \overset{(iv)}{\leq} 2\exp\left(-\frac{\kappa^2\tilde{\epsilon}^2\tau_k^\omega}{8c(c+\kappa)V_{\max}^2}\right),$$

where (i) follows from Lemma 7, (ii) follows from Lemma 14, (iii) follows because $|T^A(t_1, t_2)| \le t_2 - t_1 + 1$ and (iv) holds because

$$\tau_{k+2} - \tau_k = \frac{2c}{\kappa}\tau_{k+1}^\omega + \frac{2c}{\kappa}\tau_k^\omega = \frac{2c}{\kappa}\left(\tau_k + \frac{2c}{\kappa}\tau_k^\omega\right)^\omega + \frac{2c}{\kappa}\tau_k^\omega \le \frac{4c(c+\kappa)}{\kappa^2}\tau_k^\omega.$$

**Proof of Proposition 2**

Now we bound $\|r_t\|$ by combining the bounds of $Y_{t;\tau_k}$ and $W_{t;\tau_k}$. Applying the union bound in Lemma 8 yields

$$\mathbb{P}\left[\forall(s,a), \forall k \in [0,m], \forall t \in [\tau_{k+1}, \tau_{k+2}), |W_{t;\tau_k}(s,a)| \le \frac{\Delta}{2+\Delta}\beta D_k | E, F\right]$$

$$\ge 1 - \sum_{k=0}^m |\mathcal{S}||\mathcal{A}|(\tau_{k+2} - \tau_{k+1}) \cdot \mathbb{P}\left[|W_{t;\tau_k}(s,a)| > \frac{\Delta}{2+\Delta}\beta D_k \Big| t \in [\tau_{k+1}, \tau_{k+2}), E, F\right]$$

$$\ge 1 - \sum_{k=0}^m |\mathcal{S}||\mathcal{A}|\frac{2c}{\kappa}\tau_{k+1}^\omega \cdot 2\exp\left(-\frac{\kappa^2\left(\frac{\Delta}{2+\Delta}\right)^2\beta^2 D_k^2 \tau_k^\omega}{8c(c+\kappa)V_{\max}^2}\right)$$

$$\ge 1 - \sum_{k=0}^m |\mathcal{S}||\mathcal{A}|\frac{2c}{\kappa}\left(1 + \frac{2c}{\kappa}\right)\tau_k^\omega \cdot 2\exp\left(-\frac{\kappa^2\left(\frac{\Delta}{2+\Delta}\right)^2\beta^2 D_k^2 \tau_k^\omega}{8c(c+\kappa)V_{\max}^2}\right)$$

$$\overset{\text{(i)}}{\ge} 1 - \sum_{k=0}^m |\mathcal{S}||\mathcal{A}|\frac{2c}{\kappa}\left(1 + \frac{2c}{\kappa}\right)\tau_k^\omega \cdot 2\exp\left(-\frac{\kappa^2\left(\frac{\Delta}{2+\Delta}\right)^2\beta^2\epsilon^2 \tau_k^\omega}{8c(c+\kappa)V_{\max}^2}\right) \qquad (15)$$

$$\overset{\text{(ii)}}{\ge} 1 - \frac{4c}{\kappa}\left(1 + \frac{2c}{\kappa}\right)\sum_{k=0}^m |\mathcal{S}||\mathcal{A}| \cdot \exp\left(-\frac{\kappa^2\left(\frac{\Delta}{2+\Delta}\right)^2\beta^2\epsilon^2 \tau_k^\omega}{16c(c+\kappa)V_{\max}^2}\right)$$

$$\ge 1 - \frac{4c(m+1)}{\kappa}\left(1 + \frac{2c}{\kappa}\right)|\mathcal{S}||\mathcal{A}|\exp\left(-\frac{\kappa^2\left(\frac{\Delta}{2+\Delta}\right)^2\beta^2\epsilon^2 \tau_1^\omega}{16c(c+\kappa)V_{\max}^2}\right),$$

where (i) follows because $D_k \ge D_m \ge \epsilon$, and (ii) follows from Lemma 9 by substituting $a = \frac{16c(c+\kappa)V_{\max}^2}{\kappa^2\left(\frac{\Delta}{2+\Delta}\right)^2\beta^2\epsilon^2}, b = 1$ and observing that

$$\tau_k^\omega \ge \hat{\tau}_1^\omega \ge \frac{32c(c+\kappa)V_{\max}^2}{\kappa^2\left(\frac{\Delta}{2+\Delta}\right)^2\beta^2\epsilon^2}\ln\left(\frac{16c(c+\kappa)V_{\max}^2}{\kappa^2\left(\frac{\Delta}{2+\Delta}\right)^2\beta^2\epsilon^2}\right) = 2ab\ln ab.$$

Note that $Y_{t;\tau_k}(s,a)$ is deterministic. We complete this proof by observing that

$$\mathbb{P}\left[\forall k \in [0,m], \forall t \in [\tau_{k+1}, \tau_{k+2}), \|Q_t^A - Q^*\| \le D_{k+1}|E,F\right]$$

$$\ge \mathbb{P}\left[\forall(s,a), \forall k \in [0,m], \forall t \in [\tau_{k+1}, \tau_{k+2}), |W_{t;\tau_k}(s,a)| \le \frac{\Delta}{2+\Delta}\beta D_k|E,F\right].$$

## C.3  Part III: Bounding $\|Q_t^A - Q^*\|$

We combine the results in the first two parts, and provide a high probability bound on $\|r_t\|$ with further probabilistic arguments, which exploit the high probability bounds on $\mathbb{P}(E)$ in Proposition 1 and $\mathbb{P}(F)$ in the following lemma.

**Lemma 15.** *Let the sequence $\tau_k$ be the same as given in Lemma 10, i.e. $\tau_{k+1} = \tau_k + \frac{2c}{\kappa}\tau_k^\omega$ for $k \ge 1$. Then we have*

$$\mathbb{P}\left[\forall k \in [1,m], I_k^A \ge c\tau_k^\omega\right] \ge 1 - m\exp\left(-\frac{(1-\kappa)^2 c\tau_1^\omega}{\kappa}\right).$$

*where $I_k^A$ denotes the number of iterations updating $Q^A$ at epoch $k$.*

*Proof.* The event updating $Q^A$ is a binomial random variable. To be specific, at iteration $t$ we define

$$J_t^A = \begin{cases} 1, & \text{updating } Q^A; \\ 0, & \text{updating } Q^B. \end{cases}$$

Clearly, the events are independent across iterations. Therefore, for a given epoch $[\tau_k, \tau_{k+1})$, $I_k^A = \sum_{t=\tau_k}^{\tau_{k+1}-1} J_t^A$ is a binomial random variable satisfying the distribution $Binomial(\tau_{k+1} - \tau_k, 0.5)$. In the following, we use the tail bound of a binomial random variable. That is, if a random variable $X \sim Binomial(n, p)$, by Hoeffding's inequality we have $\mathbb{P}(X \leq x) \leq \exp\left(-\frac{2(np-x)^2}{n}\right)$ for $x < np$, which implies $\mathbb{P}(X \leq \kappa np) \leq \exp\left(-2np^2(1-\kappa)^2\right)$ for any fixed $\kappa \in (0, 1)$.

If $k = 0$, $I_0^A \sim Binomial(\tau_1, 0.5)$. Thus the tail bound yields

$$\mathbb{P}\left[I_0^A \leq \frac{\kappa}{2} \cdot \tau_1\right] \leq \exp\left(-\frac{(1-\kappa)^2 \tau_1}{2}\right).$$

If $k \geq 1$, since $\tau_{k+1} - \tau_k = \frac{2c}{\kappa}\tau_k^\omega$, we have $I_k^A \sim Binomial\left(\frac{2c}{\kappa}\tau_k^\omega, 0.5\right)$. Thus the tail bound of a binomial random variable gives

$$\mathbb{P}\left[I_k^A \leq \frac{\kappa}{2} \cdot \frac{2c}{\kappa}\tau_k^\omega\right] \leq \exp\left(-\frac{(1-\kappa)^2 c\tau_k^\omega}{\kappa}\right).$$

Then by the union bound, we have

$$\begin{aligned}
\mathbb{P}\left[\forall k \in [1, m], I_k^A \geq c\tau_k^\omega\right] &= \mathbb{P}\left[\forall k \in [1, m], I_k^A \geq \frac{\kappa}{2} \cdot \frac{2c}{\kappa}\tau_k^\omega\right] \\
&\geq 1 - \sum_{k=1}^{m} \exp\left(-\frac{(1-\kappa)^2 c\tau_k^\omega}{\kappa}\right) \\
&\geq 1 - m\exp\left(-\frac{(1-\kappa)^2 c\tau_1^\omega}{\kappa}\right).
\end{aligned}$$

$\square$

We further give the following Lemma 16 and Lemma 17 before proving Theorem 1. Lemma 16 characterizes the number of blocks to achieve $\epsilon$-accuracy given $D_k$ defined in Lemma 10.

**Lemma 16.** *Let $D_{k+1} = (1 - \beta)D_k$ with $\beta = \frac{1-\gamma}{4}$, $D_0 = \frac{2\gamma V_{\max}}{1-\gamma}$. Then for $m \geq \frac{4}{1-\gamma} \ln \frac{2\gamma V_{\max}}{\epsilon(1-\gamma)}$, we have $D_m \leq \epsilon$.*

*Proof.* By the definition of $D_k$, we have $D_k = (1 - \beta)^k D_0$. Then we obtain

$$D_k \leq \epsilon \iff (1-\beta)^k D_0 \leq \epsilon \iff \frac{1}{(1-\beta)^k} \geq \frac{D_0}{\epsilon} \iff k \geq \frac{\ln(D_0/\epsilon)}{\ln(1/(1-\beta))}.$$

Further observe that $\ln \frac{1}{1-x} \leq x$ if $x \in (0, 1)$. Thus we have

$$k \geq \frac{1}{\beta} \ln \frac{D_0}{\epsilon} = \frac{4}{1-\gamma} \ln \frac{2\gamma V_{\max}}{\epsilon(1-\gamma)}.$$

$\square$

From the above lemma, it suffices to find the starting time at epoch $m^* = \left\lceil \frac{4}{1-\gamma} \ln \frac{2\gamma V_{\max}}{\epsilon(1-\gamma)} \right\rceil$.

The next lemma is useful to calculate the total iterations given the initial epoch length and number of epochs.

**Lemma 17.** *(Even-Dar and Mansour, 2003, Lemma 32) Consider a sequence $\{x_k\}$ satisfying*

$$x_{k+1} = x_k + cx_k^\omega = x_1 + \sum_{i=1}^{k} cx_i^\omega.$$

*Then for any constant $\omega \in (0, 1)$, we have*

$$x_k = O\left((x_1^{1-\omega} + ck)^{\frac{1}{1-\omega}}\right) = O\left(x_1 + (ck)^{\frac{1}{1-\omega}}\right).$$

**Proof of Theorem 1**

Now we are ready to prove Theorem 1 based on the results obtained so far.

Let $m^* = \left\lceil \frac{4}{1-\gamma} \ln \frac{2\gamma V_{\max}}{\epsilon(1-\gamma)} \right\rceil$, then $G_{m^*-1} \geq \sigma\epsilon, D_{m^*-1} \geq \epsilon$. Thus we obtain

$$\mathbb{P}(\left\|Q^A_{\tau_{m^*}}(s,a) - Q^*\right\| \leq \epsilon)$$

$$\geq \mathbb{P}\left[\forall k \in [0, m^*-1], \forall t \in [\tau_{k+1}, \tau_{k+2}), \left\|Q^A_t - Q^*\right\| \leq D_{k+1}\right]$$

$$= \mathbb{P}\left[\forall k \in [0, m^*-1], \forall t \in [\tau_{k+1}, \tau_{k+2}), \left\|Q^A_t - Q^*\right\| \leq D_{k+1}|E, F\right] \cdot \mathbb{P}(E \cap F)$$

$$\geq \mathbb{P}\left[\forall k \in [0, m^*-1], \forall t \in [\tau_{k+1}, \tau_{k+2}), \left\|Q^A_t - Q^*\right\| \leq D_{k+1}|E, F\right]$$
$$\quad \cdot (\mathbb{P}(E) + \mathbb{P}(F) - 1)$$

$$\overset{(i)}{\geq} \mathbb{P}\left[\forall k \in [0, m^*-1], \forall t \in [\tau_{k+1}, \tau_{k+2}), \left\|Q^A_t - Q^*\right\| \leq D_{k+1}|E, F\right]$$
$$\quad \cdot \left(\mathbb{P}\left[\forall q \in [0, m^*-1], \forall t \in [\hat{\tau}_{q+1}, \hat{\tau}_{q+2}), \left\|Q^B_t - Q^A_t\right\| \leq G_{q+1}\right] + \mathbb{P}(F) - 1\right)$$

$$\overset{(ii)}{\geq} \left[1 - \frac{4cm^*}{\kappa}\left(1 + \frac{2c}{\kappa}\right)|\mathcal{S}||\mathcal{A}|\exp\left(-\frac{\kappa^2\left(\frac{\Delta}{2+\Delta}\right)^2\beta^2\epsilon^2\tau_1^\omega}{16c(c+\kappa)V_{\max}^2}\right)\right]$$

$$\cdot \left[1 - \frac{4cm^*}{\kappa}\left(1 + \frac{2c}{\kappa}\right)|\mathcal{S}||\mathcal{A}|\exp\left(-\frac{\kappa^2\left(\frac{\Delta}{2+\Delta}\right)^2\xi^2\sigma^2\epsilon^2\hat{\tau}_1^\omega}{64c(c+\kappa)V_{\max}^2}\right) - m^*\exp\left(-\frac{(1-\kappa)^2 c\hat{\tau}_1^\omega}{\kappa}\right)\right]$$

$$\geq 1 - \frac{4cm^*}{\kappa}\left(1 + \frac{2c}{\kappa}\right)|\mathcal{S}||\mathcal{A}|\exp\left(-\frac{\kappa^2\left(\frac{\Delta}{2+\Delta}\right)^2\beta^2\epsilon^2\tau_1^\omega}{16c(c+\kappa)V_{\max}^2}\right)$$

$$- \frac{4cm^*}{\kappa}\left(1 + \frac{2c}{\kappa}\right)|\mathcal{S}||\mathcal{A}|\exp\left(-\frac{\kappa^2\left(\frac{\Delta}{2+\Delta}\right)^2\xi^2\sigma^2\epsilon^2\hat{\tau}_1^\omega}{64c(c+\kappa)V_{\max}^2}\right) - m^*\exp\left(-\frac{(1-\kappa)^2 c\hat{\tau}_1^\omega}{\kappa}\right)$$

$$\overset{(iii)}{\geq} 1 - \frac{12cm^*}{\kappa}\left(1 + \frac{2c}{\kappa}\right)|\mathcal{S}||\mathcal{A}|\exp\left(-\frac{\kappa^2(1-\kappa)^2\left(\frac{\Delta}{2+\Delta}\right)^2\xi^2\sigma^2\epsilon^2\hat{\tau}_1^\omega}{64c(c+\kappa)V_{\max}^2}\right),$$

where (i) follows from Lemma 10, (ii) follows from Proposition 1 and 2 and (iii) holds due to the fact that

$$\frac{4cm^*}{\kappa}\left(1 + \frac{2c}{\kappa}\right)|\mathcal{S}||\mathcal{A}| = \max\left\{\frac{4cm^*}{\kappa}\left(1 + \frac{2c}{\kappa}\right)|\mathcal{S}||\mathcal{A}|, m^*\right\},$$

$$\frac{\kappa^2(1-\kappa)^2\left(\frac{\Delta}{2+\Delta}\right)^2\xi^2\sigma^2\epsilon^2\hat{\tau}_1^\omega}{64c(c+\kappa)V_{\max}^2} \leq \min\left\{\frac{\kappa^2\left(\frac{\Delta}{2+\Delta}\right)^2\beta^2\epsilon^2\hat{\tau}_1^\omega}{16c(c+\kappa)V_{\max}^2}, \frac{(1-\kappa)^2\hat{\tau}_1^\omega}{\kappa}, \frac{\kappa^2\left(\frac{\Delta}{2+\Delta}\right)^2\xi^2\sigma^2\epsilon^2\hat{\tau}_1^\omega}{64c(c+\kappa)V_{\max}^2}\right\}.$$

By setting

$$1 - \frac{12cm^*}{\kappa}\left(1 + \frac{2c}{\kappa}\right)|\mathcal{S}||\mathcal{A}|\exp\left(-\frac{\kappa^2(1-\kappa)^2\left(\frac{\Delta}{2+\Delta}\right)^2\xi^2\sigma^2\epsilon^2\hat{\tau}_1^\omega}{64c(c+\kappa)V_{\max}^2}\right) \geq 1 - \delta,$$

we obtain

$$\hat{\tau}_1 \geq \left( \frac{64c(c+\kappa)V_{\max}^2}{\kappa^2(1-\kappa)^2 \left(\frac{\Delta}{2+\Delta}\right)^2 \xi^2 \sigma^2 \epsilon^2} \ln \frac{12cm^*|\mathcal{S}||\mathcal{A}|(2c+\kappa)}{\kappa^2\delta} \right)^{\frac{1}{\omega}}.$$

Considering the conditions on $\hat{\tau}_1$ in Proposition 1 and Proposition 2, we choose

$$\hat{\tau}_1 = \Theta \left( \left( \frac{V_{\max}^2}{(1-\gamma)^4\epsilon^2} \ln \frac{m^*|\mathcal{S}||\mathcal{A}|V_{\max}^2}{(1-\gamma)^4\epsilon^2\delta} \right)^{\frac{1}{\omega}} \right).$$

Finally, applying the number of iterations $m^* = \left\lceil \frac{4}{1-\gamma} \ln \frac{2\gamma V_{\max}}{\epsilon(1-\gamma)} \right\rceil$ and Lemma 17, we conclude that it suffices to let

$$T = \Omega \left( \left( \frac{V_{\max}^2}{(1-\gamma)^4\epsilon^2} \ln \frac{m^*|\mathcal{S}||\mathcal{A}|V_{\max}^2}{(1-\gamma)^4\epsilon^2\delta} \right)^{\frac{1}{\omega}} + \left( \frac{2c}{\kappa} \frac{1}{1-\gamma} \ln \frac{\gamma V_{\max}}{(1-\gamma)\epsilon} \right)^{\frac{1}{1-\omega}} \right)$$

$$= \Omega \left( \left( \frac{V_{\max}^2}{(1-\gamma)^4\epsilon^2} \ln \frac{|\mathcal{S}||\mathcal{A}|V_{\max}^2 \ln(\frac{V_{\max}}{(1-\gamma)\epsilon})}{(1-\gamma)^5\epsilon^2\delta} \right)^{\frac{1}{\omega}} + \left( \frac{1}{1-\gamma} \ln \frac{V_{\max}}{(1-\gamma)\epsilon} \right)^{\frac{1}{1-\omega}} \right)$$

$$= \Omega \left( \left( \frac{V_{\max}^2}{(1-\gamma)^4\epsilon^2} \ln \frac{|\mathcal{S}||\mathcal{A}|V_{\max}^2}{(1-\gamma)^5\epsilon^2\delta} \right)^{\frac{1}{\omega}} + \left( \frac{1}{1-\gamma} \ln \frac{V_{\max}}{(1-\gamma)\epsilon} \right)^{\frac{1}{1-\omega}} \right),$$

to attain an $\epsilon$-accurate Q-estimator.

# D  Proof of Theorem 2

The main idea of this proof is similar to that of Theorem 1 with further efforts to characterize the effects of asynchronous sampling. The proof also consists of three parts: (a) Part I which analyzes the stochastic error propagation between the two Q-estimators $\left\|Q_t^B - Q_t^A\right\|$; (b) Part II which analyzes the error dynamics between one Q-estimator and the optimum $\left\|Q_t^A - Q^*\right\|$ conditioned on the error event in Part I; and (c) Part III which bounds the unconditional error $\left\|Q_t^A - Q^*\right\|$.

To proceed the proof, we first introduce the following notion of valid iterations for any fixed state-action pair $(s, a)$.

**Definition 2.** *We define $T(s, a)$ as the collection of iterations if a state-action pair $(s, a)$ is used to update the Q-function $Q^A$ or $Q^B$, and $T^A(s, a)$ as the collection of iterations specifically updating $Q^A(s, a)$. In addition, we denote $T(s, a, t_1, t_2)$ and $T^A(s, a, t_1, t_2)$ as the set of iterations updating $(s, a)$ and $Q^A(s, a)$ between time $t_1$ and $t_2$, respectively. That is,*

$$T(s, a, t_1, t_2) = \{t : t \in [t_1, t_2] \text{ and } t \in T(s, a)\},$$
$$T^A(s, a, t_1, t_2) = \{t : t \in [t_1, t_2] \text{ and } t \in T^A(s, a)\}.$$

*Correspondingly, the number of iterations updating $(s, a)$ between time $t_1$ and $t_2$ equals the cardinality of $T(s, a, t_1, t_2)$ which is denoted as $|T(s, a, t_1, t_2)|$. Similarly, the number of iterations updating $Q^A(s, a)$ between time $t_1$ and $t_2$ is denoted as $|T^A(s, a, t_1, t_2)|$.*

Given Assumption 1, we can obtain some properties of the quantities defined above.

**Lemma 18.** *It always holds that $|T(s, a, t_1, t_2)| \leq t_2 - t_1 + 1$ and $|T^A(s, a, t_1, t_2)| \leq t_2 - t_1 + 1$. In addition, suppose that Assumption 1 holds. Then we have $T(s, a, t, t + 2kL - 1) \geq k$ for any $t \geq 0$.*

*Proof.* Since in a consecutive $2L$ running iterations of Algorithm 1, either $Q^A$ or $Q^B$ is updated at least $L$ times. Then following from Assumption 1, $(s, a)$ is visited at least once for each $2L$ running iterations of Algorithm 1, which immediately implies this proposition. $\square$

Now we proceed our proof by three parts.

## D.1 Part I: Bounding $\left\| Q_t^B - Q_t^A \right\|$

We upper bound $\left\| Q_t^B - Q_t^A \right\|$ block-wisely using a decreasing sequence $\{G_q\}_{q \geq 0}$ as defined in Proposition 3 below.

**Proposition 3.** *Fix $\epsilon > 0, \kappa \in (\ln 2, 1)$ and $\Delta \in (0, e^\kappa - 2)$. Consider asynchronous double Q-learning using a polynomial learning rate $\alpha_t = \frac{1}{t^\omega}$ with $\omega \in (0,1)$. Suppose that Assumption 1 holds. Let $G_q = (1 - \xi)^q G_0$ with $G_0 = V_{\max}$ and $\xi = \frac{1-\gamma}{4}$. Let $\hat\tau_{q+1} = \hat\tau_q + \frac{2cL}{\kappa} \hat\tau_q^\omega$ for $q \geq 1$ with $c \geq \frac{L\kappa(\ln(2+\Delta)+1/\tau_1^\omega)}{2(\kappa - \ln(2+\Delta) - 1/\tau_1^\omega)}$ and $\hat\tau_1$ as the finishing time of the first block satisfying*

$$
\hat\tau_1 \geq \max \left\{ \left( \frac{1}{\kappa - \ln(2+\Delta)} \right)^{\frac{1}{\omega}}, \left( \frac{128cL(cL+\kappa)V_{\max}^2}{\kappa^2 \left( \frac{\Delta}{2+\Delta} \right)^2 \xi^2 \sigma^2 \epsilon^2} \ln \left( \frac{64cL(cL+\kappa)V_{\max}^2}{\kappa^2 \left( \frac{\Delta}{2+\Delta} \right)^2 \xi^2 \sigma^2 \epsilon^2} \right) \right)^{\frac{1}{\omega}} \right\}.
$$

*Then for any $n$ such that $G_n \geq \sigma\epsilon$, we have*

$$
\mathbb{P} \left[ \forall q \in [0,n], \forall t \in [\hat\tau_{q+1}, \hat\tau_{q+2}), \left\| Q_t^B - Q_t^A \right\| \leq G_{q+1} \right]
$$

$$
\geq 1 - \frac{4cL(n+1)}{\kappa} \left( 1 + \frac{2cL}{\kappa} \right) |\mathcal{S}||\mathcal{A}| \exp \left( - \frac{\kappa^2 \left( \frac{\Delta}{2+\Delta} \right)^2 \xi^2 \sigma^2 \epsilon^2 \hat\tau_1^\omega}{64cL(cL+\kappa)V_{\max}^2} \right).
$$

The proof of Proposition 3 consists of the following steps. Since the main idea of the proofs is similar to that of Proposition 1, we will focus on pointing out the difference. We continue to use the notation $u_t^{BA}(s,a) := Q_t^B(s,a) - Q_t^A(s,a)$.

### Step 1: Characterizing the dynamics of $u_t^{BA}$

First, we observe that when $(s,a)$ is visited at time $t$, i.e., $t \in T(s,a)$, Lemmas 2 and 3 still apply. Otherwise, $u^{BA}$ is not updated. Thus, we have

$$
u_{t+1}^{BA}(s,a) = \begin{cases} u_t^{BA}(s,a), & t \notin T(s,a); \\ (1-\alpha_t)u_t^{BA}(s,a) + \alpha_t F_t(s,a), & t \in T(s,a), \end{cases}
$$

where $F_t$ satisfies

$$
\|\mathbb{E}[F_t | \mathcal{F}_t]\| \leq \frac{1+\gamma}{2} \left\| u_t^{BA} \right\|,
$$

where the filtration $\mathcal{F}$ in the asynchronous double Q-learning case is given by $\mathcal{F}_t = \sigma(s_k, T(s,a,0,k), R_{k-1}, 2 \leq k \leq t)$.

For $t \in T(s,a)$, we rewrite the dynamics of $u_t^{BA}(s,a)$ as

$$
u_{t+1}^{BA}(s,a) = (1-\alpha_t)u_t^{BA}(s,a) + \alpha_t F_t = (1-\alpha_t)u_t^{BA}(s,a) + \alpha_t(h_t(s,a) + z_t(s,a)),
$$

where $h_t(s,a) = \mathbb{E}[F_t(s,a)|\mathcal{F}_t]$ and $z_t(s,a) = F_t(s,a) - \mathbb{E}[F_t(s,a)|\mathcal{F}_t]$.

In the following steps, we use induction to proceed the proof of Proposition 3. Given $G_q$ defined in Proposition 3, since $\left\| u_t^{BA} \right\| \leq G_0$ holds for all $t$, and thus it holds for $t \in [0, \hat\tau_1]$. Now suppose $\hat\tau_q$ satisfies that $\left\| u_t^{BA} \right\| \leq G_q$ for any $t \geq \hat\tau_q$. Then we will show there exists $\hat\tau_{q+1} = \hat\tau_q + \frac{2cL}{\kappa} \hat\tau_q^\omega$ such that $\left\| u_t^{BA} \right\| \leq G_{q+1}$ for any $t \geq \hat\tau_{q+1}$.

### Step 2: Constructing sandwich bounds

We first observe that the following sandwich bound still holds for all $t \geq \hat\tau_q$.

$$
-X_{t;\hat\tau_q}(s,a) + Z_{t;\hat\tau_q}(s,a) \leq u_t^{BA}(s,a) \leq X_{t;\hat\tau_q}(s,a) + Z_{t;\hat\tau_q}(s,a),
$$

where $Z_{t;\hat\tau_q}(s,a)$ is defined as

$$
Z_{t+1;\hat\tau_q}(s,a) = \begin{cases} Z_{t;\hat\tau_q}(s,a), & t \notin T(s,a) \\ (1-\alpha_t)Z_{t;\hat\tau_q}(s,a) + \alpha_t z_t(s,a), & t \in T(s,a), \end{cases}
$$

with the initial condition $Z_{\hat{\tau}_q;\hat{\tau}_q}(s,a) = 0$, and $X_{t;\hat{\tau}_q}(s,a)$ is defined as

$$X_{t+1;\hat{\tau}_q}(s,a) = \begin{cases} X_{t;\hat{\tau}_q}(s,a), & t \notin T(s,a) \\ (1-\alpha_t)X_{t;\hat{\tau}_q}(s,a) + \alpha_t\gamma'G_q, & t \in T(s,a), \end{cases}$$

with $X_{\hat{\tau}_q;\hat{\tau}_q}(s,a) = G_q, \gamma' = \frac{1+\gamma}{2}$.

This claim can be shown by induction. This bound clearly holds for the initial case with $t = \hat{\tau}_q$. Assume that it still holds for iteration $t$. If $t \in T(s,a)$, the proof is the same as that of Lemma 3. If $t \notin T(s,a)$, since all three sequences do not change from time $t$ to time $t+1$, the sandwich bound still holds. Thus we conclude this claim.

**Step 3: Bounding $X_{t;\hat{\tau}_q}(s,a)$**

Next, we bound the deterministic sequence $X_{t;\hat{\tau}_q}(s,a)$. Observe that $X_{t;\hat{\tau}_q}(s,a) \leq G_q$ for any $t \geq \hat{\tau}_q$. We will next show that $X_{t;\hat{\tau}_q}(s,a) \leq \left(\gamma' + \frac{2}{2+\Delta}\xi\right)G_q$ for any $t \in [\hat{\tau}_{q+1}, \hat{\tau}_{q+2})$ where $\hat{\tau}_{q+1} = \hat{\tau}_q + \frac{2cL}{\kappa}\hat{\tau}_q^\omega$.

Similarly to the proof of Lemma 5, we still rewrite $X_{\hat{\tau}_q;\hat{\tau}_q}(s,a)$ as $X_{\hat{\tau}_q;\hat{\tau}_q}(s,a) = G_q = \gamma'G_q + (1-\gamma')G_q := \gamma'G_q + \rho_{\hat{\tau}_q}$. However, in this case the dynamics of $X_{t;\hat{\tau}_q}(s,a)$ is different, which is represented as

$$X_{t+1;\hat{\tau}_q}(s,a) = \begin{cases} X_{t;\hat{\tau}_q}(s,a), & t \notin T(s,a) \\ (1-\alpha_t)X_{t;\hat{\tau}_q}(s,a) + \alpha_t\gamma'G_q = \gamma'G_q + (1-\alpha_t)\rho_t, & t \in T(s,a). \end{cases}$$

where $\rho_{t+1} = (1-\alpha_t)\rho_t$ when $t \in T(s,a)$. By the definition of $\rho_t$, we obtain

$$\rho_t = \rho_{\hat{\tau}_q}\prod_{i \in T(s,a,\hat{\tau}_q,t-1)}(1-\alpha_i) = (1-\gamma')G_q\prod_{i \in T(s,a,\hat{\tau}_q,t-1)}(1-\alpha_i)$$

$$\leq (1-\gamma')G_q\prod_{i \in T(s,a,\hat{\tau}_q,\hat{\tau}_{q+1}-1)}\left(1-\frac{1}{i^\omega}\right) \leq (1-\gamma')G_q\prod_{i=\hat{\tau}_{q+1}-|T(s,a,\hat{\tau}_q,\hat{\tau}_{q+1}-1)|}^{\hat{\tau}_{q+1}-1}\left(1-\frac{1}{i^\omega}\right)$$

$$\overset{(i)}{\leq} (1-\gamma')G_q\prod_{i=\hat{\tau}_{q+1}-\frac{c}{\kappa}\hat{\tau}_q^\omega}^{\hat{\tau}_{q+1}-1}\left(1-\frac{1}{i^\omega}\right) \overset{(ii)}{\leq} (1-\gamma')G_q\exp\left(-\frac{\frac{c}{\kappa}\hat{\tau}_q^\omega - 1}{(\hat{\tau}_{q+1}-1)^\omega}\right)$$

$$\leq (1-\gamma')G_q\exp\left(-\frac{\frac{c}{\kappa}\hat{\tau}_q^\omega - 1}{\hat{\tau}_{q+1}^\omega}\right) = (1-\gamma')G_q\exp\left(-\frac{c}{\kappa}\left(\frac{\hat{\tau}_q}{\hat{\tau}_{q+1}}\right)^\omega + \frac{1}{\hat{\tau}_{q+1}^\omega}\right)$$

$$\overset{(iii)}{\leq} (1-\gamma')G_q\exp\left(-\frac{c}{\kappa}\frac{1}{1+\frac{2cL}{\kappa}} + \frac{1}{\hat{\tau}_1^\omega}\right),$$

where (i) follows from Lemma 18, (ii) follows Lemma 4, and (iii) follows because $\hat{\tau}_q \geq \hat{\tau}_1$ and

$$\left(\frac{\hat{\tau}_q}{\hat{\tau}_{q+1}}\right)^\omega \geq \frac{\hat{\tau}_q}{\hat{\tau}_{q+1}} = \frac{\hat{\tau}_q}{\hat{\tau}_q + \frac{2cL}{\kappa}\hat{\tau}_q^\omega} \geq \frac{1}{1+\frac{2cL}{\kappa}}.$$

Since $\kappa \in (\ln 2, 1)$ and $\Delta \in (0, e^\kappa - 2)$, we have $\ln(2+\Delta) \in (0, \kappa)$. Further, observing $\hat{\tau}_1^\omega > \frac{1}{\kappa-\ln(2+\Delta)}$, we obtain $\ln(2+\Delta) + \frac{1}{\hat{\tau}_1^\omega} \in (0, \kappa)$. Last, since $c \geq \frac{L\kappa(\ln(2+\Delta)+1/\hat{\tau}_1^\omega)}{2(\kappa-\ln(2+\Delta)-1/\hat{\tau}_1^\omega)}$, we have $-\frac{c}{1+\frac{2c}{\kappa}} + \frac{1}{\hat{\tau}_1^\omega} \leq -\ln(2+\Delta)$.

Finally, combining the above observations with the fact $1 - \gamma' = 2\xi$, we conclude that for any $t \geq \hat{\tau}_{q+1} = \hat{\tau}_q + \frac{2cL}{\kappa}\hat{\tau}_q^\omega$,

$$X_{t;\hat{\tau}_q}(s,a) \leq \left(\gamma' + \frac{2}{2+\Delta}\xi\right)G_q.$$

**Step 4: Bounding $Z_{t;\hat{\tau}_q}(s,a)$**

It remains to bound the stochastic sequence $Z_{t;\hat{\tau}_q}(s,a)$ by $\frac{\Delta}{2+\Delta}\xi G_q$ at epoch $q+1$. We define an auxiliary sequence $\{Z_{t;\hat{\tau}_q}^l(s,a)\}$ (which is different from that in (9)) as:

$$Z_{t;\hat{\tau}_q}^l(s,a) = \sum_{i \in T(s,a,\hat{\tau}_q,t-1)}\alpha_i\prod_{j \in T(s,a,i+1,t-1)}(1-\alpha_j)z_i(s,a).$$

Following the same arguments as the proof of Lemma 6, we conclude that $\{Z_{t;\hat{\tau}_q}^l(s,a)\}$ is a martingale sequence and satisfies

$$|Z_{t;\hat{\tau}_q}^l(s,a) - Z_{t;\hat{\tau}_q}^{l-1}(s,a)| = \alpha_{\hat{\tau}_q+l}|z_{\hat{\tau}_q+l}(s,a)| \leq \frac{2V_{\max}}{\hat{\tau}_q^\omega}.$$

In addition, note that

$$Z_{t;\hat{\tau}_q}(s,a) = Z_{t;\hat{\tau}_q}(s,a) - Z_{\hat{\tau}_q;\hat{\tau}_q}(s,a)$$
$$= \sum_{l:\hat{\tau}_q+l-1\in T(s,a,\hat{\tau}_q,t-1)} (Z_{t;\hat{\tau}_q}^l(s,a) - Z_{t;\hat{\tau}_q}^{l-1}(s,a)) + Z_{t;\hat{\tau}_q}^{\min(T(s,a,\hat{\tau}_q,t-1))}(s,a).$$

Then we apply Azuma' inequality in Lemma 7 and obtain

$$\mathbb{P}\left[|Z_{t;\hat{\tau}_q}(s,a)| > \hat{\epsilon}|t \in [\hat{\tau}_{q+1},\hat{\tau}_{q+2})\right]$$

$$\leq 2\exp\left(\frac{-\hat{\epsilon}^2}{2\sum_{l:\hat{\tau}_q+l-1\in T(s,a,\hat{\tau}_q,t-1)} (Z_{t;\hat{\tau}_q}^l(s,a) - Z_{t;\hat{\tau}_q}^{l-1}(s,a))^2 + 2\left(Z_{t;\hat{\tau}_q}^{\min(T(s,a,\hat{\tau}_q,t-1))}(s,a)\right)^2}\right)$$

$$\leq 2\exp\left(-\frac{\hat{\epsilon}^2\hat{\tau}_q^{2\omega}}{8(|T(s,a,\hat{\tau}_q,t-1)|+1)V_{\max}^2}\right) \overset{(i)}{\leq} 2\exp\left(-\frac{\hat{\epsilon}^2\hat{\tau}_q^{2\omega}}{8(t-\hat{\tau}_q)V_{\max}^2}\right)$$

$$\leq 2\exp\left(-\frac{\hat{\epsilon}^2\hat{\tau}_q^{2\omega}}{8(\hat{\tau}_{q+2}-\hat{\tau}_q)V_{\max}^2}\right) = 2\exp\left(-\frac{\hat{\epsilon}^2\hat{\tau}_q^{2\omega}}{8\left(\frac{2cL}{\kappa}\hat{\tau}_{q+1}^\omega + \frac{2cL}{\kappa}\hat{\tau}_q^\omega\right)V_{\max}^2}\right)$$

$$= 2\exp\left(-\frac{\hat{\epsilon}^2\hat{\tau}_q^{2\omega}}{8\left(\frac{2cL}{\kappa}(\hat{\tau}_q + \frac{2cL}{\kappa}\hat{\tau}_q^\omega)^\omega + \frac{2cL}{\kappa}\hat{\tau}_q^\omega\right)V_{\max}^2}\right)$$

$$\leq 2\exp\left(-\frac{\kappa^2\hat{\epsilon}^2\hat{\tau}_q^\omega}{32cL(cL+\kappa)V_{\max}^2}\right)$$

where (i) follows from Lemma 18.

**Step 5: Taking union over all blocks**

Finally, using the union bound of Lemma 8 yields

$$\mathbb{P}\left[\forall q \in [0,n], \forall t \in [\hat{\tau}_{q+1},\hat{\tau}_{q+2}), \|Q_t^B - Q_t^A\| \leq G_{q+1}\right]$$

$$\geq \mathbb{P}\left[\forall(s,a), \forall q \in [0,n], \forall t \in [\hat{\tau}_{q+1},\hat{\tau}_{q+2}), |Z_{t;\hat{\tau}_q}(s,a)| \leq \frac{\Delta}{2+\Delta}\xi G_q\right]$$

$$\geq 1 - \sum_{q=0}^n |\mathcal{S}||\mathcal{A}|(\hat{\tau}_{q+2}-\hat{\tau}_{q+1}) \cdot \mathbb{P}\left[|Z_{t;\hat{\tau}_q}(s,a)| > \frac{\Delta}{2+\Delta}\xi G_q \Big| t \in [\hat{\tau}_{q+1},\hat{\tau}_{q+2})\right]$$

$$\geq 1 - \sum_{q=0}^n |\mathcal{S}||\mathcal{A}|\frac{2cL}{\kappa}\hat{\tau}_{q+1}^\omega \cdot 2\exp\left(-\frac{\kappa^2\left(\frac{\Delta}{2+\Delta}\right)^2\xi^2 G_q^2\hat{\tau}_q^\omega}{32cL(cL+\kappa)V_{\max}^2}\right)$$

$$\geq 1 - \sum_{q=0}^n |\mathcal{S}||\mathcal{A}|\frac{2cL}{\kappa}\left(1 + \frac{2cL}{\kappa}\right)\hat{\tau}_q^\omega \cdot 2\exp\left(-\frac{\kappa^2\left(\frac{\Delta}{2+\Delta}\right)^2\xi^2 G_q^2\hat{\tau}_q^\omega}{32cL(cL+\kappa)V_{\max}^2}\right)$$

$$\overset{(i)}{\geq} 1 - \sum_{q=0}^n |\mathcal{S}||\mathcal{A}|\frac{2cL}{\kappa}\left(1 + \frac{2cL}{\kappa}\right)\hat{\tau}_q^\omega \cdot 2\exp\left(-\frac{\kappa^2\left(\frac{\Delta}{2+\Delta}\right)^2\xi^2\sigma^2\epsilon^2\hat{\tau}_q^\omega}{32cL(cL+\kappa)V_{\max}^2}\right)$$

$$\overset{(ii)}{\geq} 1 - \frac{4cL}{\kappa}\left(1 + \frac{2cL}{\kappa}\right)\sum_{q=0}^n |\mathcal{S}||\mathcal{A}| \cdot \exp\left(-\frac{\kappa^2\left(\frac{\Delta}{2+\Delta}\right)^2\xi^2\sigma^2\epsilon^2\hat{\tau}_q^\omega}{64cL(cL+\kappa)V_{\max}^2}\right)$$

$$\overset{\text{(iii)}}{\geq} 1 - \frac{4cL(n+1)}{\kappa}\left(1+\frac{2cL}{\kappa}\right)|\mathcal{S}||\mathcal{A}|\exp\left(-\frac{\kappa^2\left(\frac{\Delta}{2+\Delta}\right)^2\xi^2\sigma^2\epsilon^2\hat{\tau}_1^\omega}{64cL(cL+\kappa)V_{\max}^2}\right),$$

where (i) follows from $G_q \geq G_n \geq \sigma\epsilon$, (ii) follows from Lemma 9 by substituting $a = \frac{64cL(cL+\kappa)V_{\max}^2}{\kappa^2\left(\frac{\Delta}{2+\Delta}\right)^2\xi^2\sigma^2\epsilon^2}, b = 1$ and observing that

$$\hat{\tau}_q^\omega \geq \hat{\tau}_1^\omega \geq \frac{128cL(cL+\kappa)V_{\max}^2}{\kappa^2\left(\frac{\Delta}{2+\Delta}\right)^2\xi^2\sigma^2\epsilon^2}\ln\left(\frac{64cL(cL+\kappa)V_{\max}^2}{\kappa^2\left(\frac{\Delta}{2+\Delta}\right)^2\xi^2\sigma^2\epsilon^2}\right) = 2ab\ln ab,$$

and (iii) follows from $\hat{\tau}_q \geq \hat{\tau}_1$.

## D.2    Part II: Conditionally bounding $\left\|Q_t^A - Q^*\right\|$

We upper bound $\left\|Q_t^A - Q^*\right\|$ block-wisely by a decreasing sequence $\{D_k\}_{k\geq 0}$ conditioned on the following two events: fix a positive integer $m$,

$$G = \left\{\forall(s,a), \forall k \in [0,m], \forall t \in [\tau_{k+1}, \tau_{k+2}), \left\|Q_t^B - Q_t^A\right\| \leq \sigma D_{k+1}\right\}, \tag{16}$$
$$H = \{\forall k \in [1, m+1], I_k^A \geq cL\tau_k^\omega\}, \tag{17}$$

where $I_k^A$ denotes the number of iterations updating $Q^A$ at epoch $k$, $\tau_k$ is the starting iteration index of the $k+1$th block, and $\omega$ is the parameter of the polynomial learning rate. Roughly, Event $G$ requires that the difference between the two Q-function estimators are bounded appropriately, and Event $H$ requires that $Q^A$ is sufficiently updated in each epoch. Again, we will design $\{D_k\}_{k\geq 0}$ in a way such that the occurrence of Event $G$ can be implied from the event that $\left\|u_t^{BA}\right\|$ is bounded by $\{G_q\}_{q\geq 0}$ (see Lemma 19 below). A lower bound of the probability for Event $H$ to hold is characterized in Lemma 15 in Part III.

**Proposition 4.** *Fix $\epsilon > 0, \kappa \in (\ln 2, 1)$ and $\Delta \in (0, e^\kappa - 2)$. Consider asynchronous double Q-learning using a polynomial learning rate $\alpha_t = \frac{1}{t^\omega}$ with $\omega \in (0,1)$. Let $\{G_q\}, \{\hat{\tau}_q\}$ be as defined in Proposition 3. Define $D_k = (1-\beta)^k \frac{V_{\max}}{\sigma}$ with $\beta = \frac{1-\gamma(1+\sigma)}{2}$ and $\sigma = \frac{1-\gamma}{2\gamma}$. Let $\tau_k = \hat{\tau}_k$ for $k \geq 0$. Suppose that $c \geq \frac{L(\ln(2+\Delta)+1/\tau_1^\omega)}{2(\kappa-\ln(2+\Delta)-1/\tau_1^\omega)}$ and $\tau_1 = \hat{\tau}_1$ as the finishing time of the first epoch satisfies*

$$\tau_1 \geq \max\left\{\left(\frac{1}{\kappa-\ln(2+\Delta)}\right)^{\frac{1}{\omega}}, \left(\frac{32cL(cL+\kappa)V_{\max}^2}{\kappa^2\left(\frac{\Delta}{2+\Delta}\right)^2\beta^2\epsilon^2}\ln\left(\frac{16cL(cL+\kappa)V_{\max}^2}{\kappa^2\left(\frac{\Delta}{2+\Delta}\right)^2\beta^2\epsilon^2}\right)\right)^{\frac{1}{\omega}}\right\}.$$

*Then for any $m$ such that $D_m \geq \epsilon$, we have*

$$\mathbb{P}\left[\forall k \in [0,m], \forall t \in [\tau_{k+1}, \tau_{k+2}), \left\|Q_t^A - Q^*\right\| \leq D_{k+1}|G, H\right]$$
$$\geq 1 - \frac{4cL(m+1)}{\kappa}\left(1+\frac{2cL}{\kappa}\right)|\mathcal{S}||\mathcal{A}|\exp\left(-\frac{\kappa^2\left(1-\frac{2}{e}\right)^2\beta^2\epsilon^2\tau_1^\omega}{16cL(cL+\kappa)V_{\max}^2}\right).$$

Recall that in the proof of Proposition 2, $Q^A$ is not updated at each iteration and thus we introduced notations $T^A$ and $T^A(t_1, t_2)$ in Definition 1 to capture the convergence of the error $\left\|Q^A - Q^*\right\|$. In this proof, the only difference is that when choosing to update $Q^A$, only one $(s,a)$-pair is visited. Therefore, the proof of Proposition 4 is similar to that of Proposition 2, where most of the arguments simply substitute $T^A, T^A(t_1, t_2)$ in the proof of Proposition 2 by $T^A(s,a), T^A(s,a,t_1,t_2)$ in Definition 2, respectively. Certain bounds are affected by such substitutions. In the following, we proceed the proof of Proposition 4 in five steps, and focus on pointing out the difference from the proof of Proposition 2. More details can be referred to Appendix C.2.

**Step 1: Coupling $\{D_k\}_{k\geq 0}$ and $\{G_q\}_{q\geq 0}$**

We establish the relationship between $\{D_k\}_{k\geq 0}$ and $\{G_q\}_{q\geq 0}$ in the same way as Lemma 10. For the convenience of reference, we restate Lemma 10 in the following.

**Lemma 19.** *Let $\{G_q\}$ be defined in Proposition 3, and let $D_k = (1-\beta)^k \frac{V_{\max}}{\sigma}$ with $\beta = \frac{1-\gamma(1+\sigma)}{2}$ and $\sigma = \frac{1-\gamma}{2\gamma}$. Then we have*

$$\mathbb{P}\left[\forall(s,a), \forall q \in [0,m], \forall t \in [\hat{\tau}_{q+1}, \hat{\tau}_{q+2}), \|Q_t^B - Q_t^A\| \le G_{q+1}\right]$$
$$\le \mathbb{P}\left[\forall(s,a), \forall k \in [0,m], \forall t \in [\tau_{k+1}, \tau_{k+2}), \|Q_t^B - Q_t^A\| \le \sigma D_{k+1}\right],$$

*given that $\tau_k = \hat{\tau}_k$.*

### Step 2: Constructing sandwich bounds

Let $r_t(s,a) = Q^A(s,a) - Q^*(s,a)$ and $\tau_k$ be such that $\|r_t\| \le D_k$ for all $t \ge \tau_k$. The requirement of Event $G$ yields

$$-Y_{t;\tau_k}(s,a) + W_{t;\tau_k}(s,a) \le r_t(s,a) \le Y_{t;\tau_k}(s,a) + W_{t;\tau_k}(s,a),$$

where $W_{t;\tau_k}(s,a)$ is defined as

$$W_{t+1;\tau_k}(s,a) = \begin{cases} W_{t;\tau_k}(s,a), & t \notin T^A(s,a); \\ (1-\alpha_t)W_{t;\tau_k}(s,a) + \alpha_t w_t(s,a), & t \in T^A(s,a), \end{cases}$$

with $w_t(s,a) = \mathcal{T}_t Q_t^A(s,a) - \mathcal{T} Q_t^A(s,a)$ and $W_{\tau_k;\tau_k}(s,a) = 0$, and $Y_{t;\tau_k}(s,a)$ is given by

$$Y_{t+1;\tau_k}(s,a) = \begin{cases} Y_{t;\tau_k}(s,a), & t \notin T^A(s,a); \\ (1-\alpha_t)Y_{t;\tau_k}(s,a) + \alpha_t \gamma'' D_k, & t \in T^A(s,a), \end{cases}$$

with $Y_{\tau_k;\tau_k}(s,a) = D_k$ and $\gamma'' = \gamma(1+\sigma)$.

### Step 3: Bounding $Y_{t;\tau_k}(s,a)$

Next, we first bound $Y_{t;\tau_k}(s,a)$. Observe that $Y_{t;\tau_k}(s,a) \le D_k$ for any $t \ge \tau_k$. We will bound $Y_{t;\tau_k}(s,a)$ by $\left(\gamma'' + \frac{2}{2+\Delta}\beta\right)D_k$ for block $k+1$.

We use a similar representation of $Y_{t;\tau_k}(s,a)$ as in the proof of Lemma 13, which is given by

$$Y_{t+1;\tau_k}(s,a) = \begin{cases} Y_{t;\tau_k}(s,a), & t \notin T^A(s,a) \\ (1-\alpha_t)Y_{t;\tau_k}(s,a) + \alpha_t \gamma'' G_q = \gamma'' G_q + (1-\alpha_t)\rho_t, & t \in T^A(s,a) \end{cases}$$

where $\rho_{t+1} = (1-\alpha_t)\rho_t$ for $t \in T^A(s,a)$. By the definition of $\rho_t$, we obtain

$$\rho_t = \rho_{\tau_k} \prod_{i \in T^A(s,a,\tau_k,t-1)} (1-\alpha_i) = (1-\gamma'')D_k \prod_{i \in T^A(s,a,\tau_k,t-1)} (1-\alpha_i)$$

$$= (1-\gamma'')D_k \prod_{i \in T^A(s,a,\tau_k,t-1)} \left(1 - \frac{1}{i^\omega}\right) \overset{(i)}{\le} (1-\gamma'')D_k \prod_{i \in T^A(s,a,\tau_k,\tau_{k+1}-1)} \left(1 - \frac{1}{i^\omega}\right)$$

$$\overset{(ii)}{\le} (1-\gamma'')D_k \prod_{i=\tau_{k+1}-c\tau_k^\omega}^{\tau_{k+1}-1} \left(1 - \frac{1}{i^\omega}\right) \overset{(iii)}{\le} (1-\gamma'')D_k \exp\left(-\frac{c\tau_k^\omega - 1}{(\tau_{k+1}-1)^\omega}\right)$$

$$\le (1-\gamma'')D_k \exp\left(-\frac{c\tau_k^\omega - 1}{\tau_{k+1}^\omega}\right) = (1-\gamma'')D_k \exp\left(-c\left(\frac{\tau_k}{\tau_{k+1}}\right)^\omega + \frac{1}{\tau_{k+1}^\omega}\right)$$

$$\overset{(iv)}{\le} (1-\gamma'')D_k \exp\left(-\frac{c}{1+\frac{2Lc}{\kappa}} + \frac{1}{\tau_1^\omega}\right),$$

where (i) follows because $\alpha_i < 1$ and $t \ge \tau_{k+1}$, (ii) follows from Proposition 18 and the requirement of event $H$, (iii) follows from Lemma 9, and (iv) holds because $\tau + k \ge \tau_1$ and

$$\left(\frac{\tau_k}{\tau_{k+1}}\right)^\omega \ge \frac{\tau_k}{\tau_{k+1}} = \frac{\tau_k}{\tau_k + \frac{2cL}{\kappa}\tau_k^\omega} \ge \frac{1}{1+\frac{2cL}{\kappa}}.$$

Since $\kappa \in (\ln 2, 1)$ and $\Delta \in (0, e^\kappa - 2)$, we have $\ln(2+\Delta) \in (0, \kappa)$. Further, observing $\hat{\tau}_1^\omega > \frac{1}{\kappa - \ln(2+\Delta)}$, we obtain $\ln(2+\Delta) + \frac{1}{\hat{\tau}_1^\omega} \in (0, \kappa)$. Last, since $c \ge \frac{L(\ln(2+\Delta)+1/\hat{\tau}_1^\omega)}{2(\kappa-\ln(2+\Delta)-1/\hat{\tau}_1^\omega)}$, we have $-\frac{c}{1+\frac{2c}{\kappa}} + \frac{1}{\hat{\tau}_1^\omega} \le -\ln(2+\Delta)$.

Then, we have $\rho_t \le \frac{1-\gamma''}{2+\Delta}D_k$. Thus we conclude that for any $t \in [\tau_{k+1}, \tau_{k+2}]$,

$$Y_{t;\tau_k}(s,a) \le \left(\gamma'' + \frac{2}{2+\Delta}\beta\right)D_k.$$

**Step 4: Bounding $W_{t;\tau_k}(s,a)$**

It remains to bound $|W_{t;\tau_k}(s,a)| \le \left(1 - \frac{2}{2+\Delta}\right)\beta D_k$ for $t \in [\tau_{k+1}, \tau_{k+2})$.

Similarly to Appendix C.2.4, we define a new sequence $\{W^l_{t;\tau_k}(s,a)\}$ as

$$W^l_{t;\tau_k}(s,a) = \sum_{i \in T^A(s,a,\tau_k,\tau_k+l)} \alpha_i \underset{j \in T^A(s,a,i+1,t-1)}{\Pi}(1-\alpha_j)w_i(s,a).$$

The same arguments as the proof of Lemma 14 yields

$$|W^l_{t;\tau_k}(s,a) - W^{l-1}_{t;\tau_k}(s,a)| \le \frac{V_{\max}}{\tau_k^\omega}.$$

If we fix $\tilde\epsilon > 0$, then for any $t \in [\tau_{k+1}, \tau_{k+2})$ we have
$$\mathbb{P}\left[|W_{t;\tau_k}(s,a)| > \tilde\epsilon | t \in [\tau_{k+1}, \tau_{k+2}), G, H\right]$$

$$\le 2\exp\left(\frac{-\tilde\epsilon^2}{2\sum_{l:\tau_k+l-1 \in T^A(s,a,\tau_k,t-1)}\left(W^l_{t;\tau_k}(s,a)-W^{l-1}_{t;\tau_k}(s,a)\right)^2 + 2(W^{\min(T^A(s,a,\tau_k,t-1))}_{t;\tau_k}(s,a))^2}\right)$$

$$\le 2\exp\left(-\frac{\tilde\epsilon^2\tau_k^{2\omega}}{2(|T^A(s,a,\tau_k,t-1)|+1)V_{\max}^2}\right) \overset{(i)}{\le} 2\exp\left(-\frac{\tilde\epsilon^2\tau_k^{2\omega}}{2(t-\tau_k)V_{\max}^2}\right)$$

$$\le 2\exp\left(-\frac{\tilde\epsilon^2\tau_k^{2\omega}}{2(\tau_{k+2}-\tau_k)V_{\max}^2}\right) \overset{(ii)}{\le} 2\exp\left(-\frac{\kappa^2\tilde\epsilon^2\tau_k^\omega}{8cL(cL+\kappa)V_{\max}^2}\right)$$

$$= 2\exp\left(-\frac{\kappa^2\tilde\epsilon^2\tau_k^\omega}{8cL(cL+\kappa)V_{\max}^2}\right),$$

where (i) follows from Proposition 18 and (ii) holds because

$$\tau_{k+2}-\tau_k = \frac{2cL}{\kappa}\tau_{k+1}^\omega + \frac{2cL}{\kappa}\tau_k^\omega = \frac{2cL}{\kappa}\left(\tau_k + \frac{2cL}{\kappa}\tau_k^\omega\right)^\omega + \frac{2cL}{\kappa}\tau_k^\omega \le \frac{4cL(cL+\kappa)}{\kappa^2}\tau_k^\omega.$$

**Step 5: Taking union over all blocks**

Applying the union bound in Lemma 8, we obtain
$$\mathbb{P}\left[\forall k \in [0,m], \forall t \in [\tau_{k+1}, \tau_{k+2}), \|Q^A_t - Q^*\| \le D_{k+1}|G, H\right]$$

$$\ge \mathbb{P}\left[\forall(s,a), \forall k \in [0,m], \forall t \in [\tau_{k+1}, \tau_{k+2}), |W_{t;\tau_k}(s,a)| \le \frac{\Delta}{2+\Delta}\beta D_k|G, H\right]$$

$$\ge 1 - \sum_{k=0}^m |\mathcal{S}||\mathcal{A}|(\tau_{k+2}-\tau_{k+1}) \cdot \mathbb{P}\left[|W_{t;\tau_k}(s,a)| > \frac{\Delta}{2+\Delta}\beta D_k\Big| t \in [\tau_{k+1}, \tau_{k+2}), G, H\right]$$

$$\ge 1 - \sum_{k=0}^m |\mathcal{S}||\mathcal{A}|\frac{2cL}{\kappa}\tau_{k+1}^\omega \cdot 2\exp\left(-\frac{\kappa^2\left(\frac{\Delta}{2+\Delta}\right)^2\beta^2 D_k^2\tau_k^\omega}{8cL(cL+\kappa)V_{\max}^2}\right)$$

$$\ge 1 - \sum_{k=0}^m |\mathcal{S}||\mathcal{A}|\frac{2cL}{\kappa}\left(1+\frac{2cL}{\kappa}\right)\tau_k^\omega \cdot 2\exp\left(-\frac{\kappa^2\left(\frac{\Delta}{2+\Delta}\right)^2\beta^2 D_k^2\tau_k^\omega}{8cL(cL+\kappa)V_{\max}^2}\right)$$

$$\overset{(i)}{\ge} 1 - \sum_{k=0}^m |\mathcal{S}||\mathcal{A}|\frac{2cL}{\kappa}\left(1+\frac{2cL}{\kappa}\right)\tau_k^\omega \cdot 2\exp\left(-\frac{\kappa^2\left(\frac{\Delta}{2+\Delta}\right)^2\beta^2\epsilon^2\tau_k^\omega}{8cL(cL+\kappa)V_{\max}^2}\right)$$

$$\overset{(ii)}{\geq} 1 - \frac{4cL}{\kappa}\left(1 + \frac{2cL}{\kappa}\right)\sum_{k=0}^{m}|\mathcal{S}||\mathcal{A}| \cdot \exp\left(-\frac{\kappa^2\left(\frac{\Delta}{2+\Delta}\right)^2\beta^2\epsilon^2\tau_k^\omega}{16cL(cL+\kappa)V_{\max}^2}\right)$$

$$\geq 1 - \frac{4cL(m+1)}{\kappa}\left(1 + \frac{2cL}{\kappa}\right)|\mathcal{S}||\mathcal{A}|\exp\left(-\frac{\kappa^2\left(\frac{\Delta}{2+\Delta}\right)^2\beta^2\epsilon^2\tau_1^\omega}{16cL(cL+\kappa)V_{\max}^2}\right),$$

where (i) follows because $D_k \geq D_m \geq \epsilon$, and (ii) follows from Lemma 9 by substituting $a = \frac{16cL(cL+\kappa)V_{\max}^2}{\kappa^2\left(\frac{\Delta}{2+\Delta}\right)^2\beta^2\epsilon^2}$, $b = 1$ and observing that

$$\tau_k^\omega \geq \hat{\tau}_1^\omega \geq \frac{32cL(cL+\kappa)V_{\max}^2}{\kappa^2\left(\frac{\Delta}{2+\Delta}\right)^2\beta^2\epsilon^2}\ln\left(\frac{64cL(cL+\kappa)V_{\max}^2}{\kappa^2\left(\frac{\Delta}{2+\Delta}\right)^2\beta^2\epsilon^2}\right) = 2ab\ln ab.$$

### D.3    Part III: Bound $\left\|Q_t^A - Q^*\right\|$

In order to obtain the unconditional high-probability bound on $\left\|Q_t^A - Q^*\right\|$, we first characterize a lower bound on the probability of Event $H$. Note that the probability of Event $G$ is lower bounded in Proposition 3.

**Lemma 20.** *Let the sequence $\tau_k$ be the same as given in Lemma 19, i.e. $\tau_{k+1} = \tau_k + \frac{2cL}{\kappa}\tau_k^\omega$ for $k \geq 1$. Define $I_k^A$ as the number of iterations updating $Q^A$ at epoch $k$. Then we have*

$$\mathbb{P}\left[\forall k \in [1,m], I_k^A \geq cL\tau_k^\omega\right] \geq 1 - m\exp\left(-\frac{(1-\kappa)^2 cL\tau_1^\omega}{\kappa}\right).$$

*Proof.* We use the same idea as the proof of Lemma 15. Since we only focus on the blocks with $k \geq 1$, $I_k^A \sim Binomial\left(\frac{2cL}{\kappa}\tau_k^\omega, 0.5\right)$ in such a case. Thus the tail bound of a binomial random variable gives

$$\mathbb{P}\left[I_k^A \leq \frac{\kappa}{2} \cdot \frac{2cL}{\kappa}\tau_k^\omega\right] \leq \exp\left(-\frac{(1-\kappa)^2 cL\tau_k^\omega}{\kappa}\right).$$

Then by the union bound, we have

$$\begin{aligned}
\mathbb{P}\left[\forall k \in [1,m], I_k^A \geq cL\tau_k^\omega\right] &= \mathbb{P}\left[\forall k \in [1,m], I_k^A \geq \frac{\kappa}{2} \cdot \frac{2cL}{\kappa}\tau_k^\omega\right] \\
&\geq 1 - \sum_{k=1}^{m}\exp\left(-\frac{(1-\kappa)^2 cL\tau_k^\omega}{\kappa}\right) \\
&\geq 1 - m\exp\left(-\frac{(1-\kappa)^2 cL\tau_1^\omega}{\kappa}\right).
\end{aligned}$$

$\square$

Following from Lemma 16, it suffices to determine the starting time at epoch $m^* = \left\lceil\frac{4}{1-\gamma}\ln\frac{2\gamma V_{\max}}{\epsilon(1-\gamma)}\right\rceil$. This can be done by using Lemma 17 if we have $\hat{\tau}_1$.

Now we are ready to prove the main result of Theorem 2. By the definition of $m^*$, we know $D_{m^*-1} \geq \epsilon, G_{m^*-1} \geq \sigma\epsilon$. Then we obtain

$$\begin{aligned}
&\mathbb{P}(\left\|Q_{\tau_{m^*}}^A - Q^*\right\| \leq \epsilon) \\
&\geq \mathbb{P}\left[\forall k \in [0, m^*-1], \forall t \in [\tau_{k+1}, \tau_{k+2}), \left\|Q_t^A - Q^*\right\| \leq D_{k+1}\right] \\
&= \mathbb{P}\left[\forall k \in [0, m^*-1], \forall t \in [\tau_{k+1}, \tau_{k+2}), \left\|Q_t^A - Q^*\right\| \leq D_{k+1}|G, H\right] \cdot \mathbb{P}(G \cap H) \\
&\geq \mathbb{P}\left[\forall k \in [0, m^*-1], \forall t \in [\tau_{k+1}, \tau_{k+2}), \left\|Q_t^A - Q^*\right\| \leq D_{k+1}|G, H\right] \\
&\quad \cdot (\mathbb{P}(G) + \mathbb{P}(H) - 1)
\end{aligned}$$

$$\overset{(i)}{\geq} \mathbb{P}\left[\forall k \in [0, m^*-1], \forall t \in [\tau_{k+1}, \tau_{k+2}), \left\|Q_t^A - Q^*\right\| \leq D_{k+1}\big|G, H\right]$$
$$\cdot \left(\mathbb{P}\left[\forall q \in [0, m^*-1], \forall t \in [\hat{\tau}_{q+1}, \hat{\tau}_{q+2}), \left\|Q_t^B - Q_t^A\right\| \leq G_{q+1}\right] + \mathbb{P}(H) - 1\right)$$

$$\overset{(ii)}{\geq} \left[1 - \frac{4cLm^*}{\kappa}\left(1 + \frac{2cL}{\kappa}\right)|\mathcal{S}||\mathcal{A}|\exp\left(-\frac{\kappa^2\left(\frac{\Delta}{2+\Delta}\right)^2\beta^2\epsilon^2\tau_1^\omega}{16cL(cL+\kappa)V_{\max}^2}\right)\right]$$

$$\cdot \left[1 - \frac{4cLm^*}{\kappa}\left(1 + \frac{2cL}{\kappa}\right)|\mathcal{S}||\mathcal{A}|\exp\left(-\frac{\kappa^2\left(\frac{\Delta}{2+\Delta}\right)^2\xi^2\sigma^2\epsilon^2\hat{\tau}_1^\omega}{64cL(cL+\kappa)V_{\max}^2}\right)\right.$$

$$\left. -m^*\exp\left(-\frac{(1-\kappa)^2cL\hat{\tau}_1^\omega}{\kappa}\right)\right]$$

$$\geq 1 - \frac{4cLm^*}{\kappa}\left(1 + \frac{2cL}{\kappa}\right)|\mathcal{S}||\mathcal{A}|\exp\left(-\frac{\kappa^2\left(\frac{\Delta}{2+\Delta}\right)^2\beta^2\epsilon^2\tau_1^\omega}{16cL(cL+\kappa)V_{\max}^2}\right)$$

$$- \frac{4cLm^*}{\kappa}\left(1 + \frac{2cL}{\kappa}\right)|\mathcal{S}||\mathcal{A}|\exp\left(-\frac{\kappa^2\left(\frac{\Delta}{2+\Delta}\right)^2\xi^2\sigma^2\epsilon^2\hat{\tau}_1^\omega}{64cL(cL+\kappa)V_{\max}^2}\right) - m^*\exp\left(-\frac{(1-\kappa)^2cL\hat{\tau}_1^\omega}{\kappa}\right)$$

$$\overset{(iii)}{\geq} 1 - \frac{12cLm^*}{\kappa}\left(1 + \frac{2cL}{\kappa}\right)|\mathcal{S}||\mathcal{A}|\exp\left(-\frac{\kappa^2(1-\kappa)^2\left(\frac{\Delta}{2+\Delta}\right)^2\xi^2\sigma^2\epsilon^2\hat{\tau}_1^\omega}{64cL(cL+\kappa)V_{\max}^2}\right),$$

where (i) follows from Lemma 19, (ii) follows from Propositions 3 and 4 and (iii) holds due to the fact that

$$\frac{4cLm^*}{\kappa}\left(1 + \frac{2cL}{\kappa}\right)|\mathcal{S}||\mathcal{A}| = \max\left\{\frac{4cLm^*}{\kappa}\left(1 + \frac{2cL}{\kappa}\right)|\mathcal{S}||\mathcal{A}|, m^*\right\},$$

$$\frac{\kappa^2(1-\kappa)^2\left(\frac{\Delta}{2+\Delta}\right)^2\xi^2\sigma^2\epsilon^2\hat{\tau}_1^\omega}{64cL(cL+\kappa)V_{\max}^2} \leq \min\left\{\frac{\kappa^2\left(\frac{\Delta}{2+\Delta}\right)^2\beta^2\epsilon^2\hat{\tau}_1^\omega}{16cL(cL+\kappa)V_{\max}^2}, \frac{(1-\kappa)^2\hat{\tau}_1^\omega}{\kappa}, \frac{\kappa^2\left(\frac{\Delta}{2+\Delta}\right)^2\xi^2\sigma^2\epsilon^2\hat{\tau}_1^\omega}{64cL(cL+\kappa)V_{\max}^2}\right\}.$$

By setting

$$1 - \frac{12cLm^*}{\kappa}\left(1 + \frac{2cL}{\kappa}\right)|\mathcal{S}||\mathcal{A}|\exp\left(-\frac{\kappa^2(1-\kappa)^2\left(\frac{\Delta}{2+\Delta}\right)^2\xi^2\sigma^2\epsilon^2\hat{\tau}_1^\omega}{64cL(cL+\kappa)V_{\max}^2}\right) \geq 1 - \delta,$$

we obtain

$$\hat{\tau}_1 \geq \left(\frac{64cL(cL+\kappa)V_{\max}^2}{\kappa^2(1-\kappa)^2\left(\frac{\Delta}{2+\Delta}\right)^2\xi^2\sigma^2\epsilon^2}\ln\frac{12m^*|\mathcal{S}||\mathcal{A}|cL(2cL+\kappa)}{\kappa^2\delta}\right)^{\frac{1}{\omega}}.$$

Combining with the requirement of $\hat{\tau}_1$ in Propositions 3 and 4, we can choose

$$\hat{\tau}_1 = \Theta\left(\left(\frac{L^4V_{\max}^2}{(1-\gamma)^4\epsilon^2}\ln\frac{m^*|\mathcal{S}||\mathcal{A}|L^4V_{\max}^2}{(1-\gamma)^4\epsilon^2\delta}\right)^{\frac{1}{\omega}}\right).$$

Finally, applying $m^* = \left\lceil\frac{4}{1-\gamma}\ln\frac{2\gamma V_{\max}}{\epsilon(1-\gamma)}\right\rceil$ and Lemma 17, we conclude that it suffices to let

$$T = \Omega\left(\left(\frac{L^4V_{\max}^2}{(1-\gamma)^4\epsilon^2}\ln\frac{m^*|\mathcal{S}||\mathcal{A}|L^4V_{\max}^2}{(1-\gamma)^4\epsilon^2\delta}\right)^{\frac{1}{\omega}} + \left(\frac{2cL}{\kappa}\frac{1}{1-\gamma}\ln\frac{\gamma V_{\max}}{(1-\gamma)\epsilon}\right)^{\frac{1}{1-\omega}}\right)$$

$$= \Omega \left( \left( \frac{L^4 V_{\max}^2}{(1-\gamma)^4 \epsilon^2} \ln \frac{|\mathcal{S}||\mathcal{A}| L^4 V_{\max}^2 \ln \frac{2\gamma V_{\max}}{\epsilon(1-\gamma)}}{(1-\gamma)^5 \epsilon^2 \delta} \right)^{\frac{1}{\omega}} + \left( \frac{2cL}{\kappa} \frac{1}{1-\gamma} \ln \frac{\gamma V_{\max}}{(1-\gamma)\epsilon} \right)^{\frac{1}{1-\omega}} \right)$$

$$= \Omega \left( \left( \frac{L^4 V_{\max}^2}{(1-\gamma)^4 \epsilon^2} \ln \frac{|\mathcal{S}||\mathcal{A}| L^4 V_{\max}^2}{(1-\gamma)^5 \epsilon^2 \delta} \right)^{\frac{1}{\omega}} + \left( \frac{L^2}{1-\gamma} \ln \frac{\gamma V_{\max}}{(1-\gamma)\epsilon} \right)^{\frac{1}{1-\omega}} \right).$$

to attain an $\epsilon$-accurate Q-estimator.