[Reviews · NeurIPS 2020]

Review 1

Summary and Contributions: Double Q-learning algorithm empirically outperforms the classical Q-learning yet there are very few theoretical garantees for double Q-learning due to the fact that it is based on two interconnected stochastic processes which prevents direct application of the techniques developped to analyse classical Q-learning. So far there are only asymptotic convergence results and this paper provides first finite-time analysis of double Q-learning.

Strengths: This is a result of high relevance for the NeurIPS comunity, it provides first elements for better understanding of the empirical success of double Q-learning over classical Q-learning. The paper provides new approach to analysing RL algorithms that are based on interconnected stochastic processes, which may inspire new theoretical results for other algorithms than double Q-learning.

Weaknesses: The results only hold for the tabular double Q-learning and it is definitely not straightforward to extend the analysis to the function approximation setting, actually used in practice. So there is still a huge effort before fully understanding empirical success of deep double Q-learning.

Correctness: The paper is technically sound. However it is definitelly not stand-alone, a detailed knowledge of Even-Dar and Mansour (2003) is assumed. This is in particular the case of Lemma 1. It would be useful to provide proof of Lemma 1 in the supplementary material. What is Rmax, a maximum value of the reward function R?

Clarity: The ideas are clear, but there are some presentation issues. For example, Algorithm 1 is supposed to be the synchronous implementation, yet in lines (8) and (11) only one state action pair is updated, which is rather asynchronous implementation. If you prefer to keep asynchronous version, please specify how do you "choose action a" (i.e. what are the assumptions on the exploration policy; for example you may want to provide a forward reference to Assumption 1 and an example of an exploration policy that verifies the assumptions). Also giving the formal definition of "converge" in Algorithm 1 would increase the readability (epsilon distance in sup metric?)

Relation to Prior Work: Yes

Reproducibility: Yes

Additional Feedback: Could you provide a discussion on the main difficulties of analysing double Q-learning in the function approximation setting? For example in the special case of linear function approximation. The discussion on the exploration policy in the asynchronous version is confusing: Assumption 1 (existance of the covering number L) suggests a deterministic exploration policy, yet there is a claim on line 262 that L >> |S||A|; why not = ? Can Assumption 1 is replaced by its "with high probability" equivalent? In Step I of Part I of the proof of Theorem 1 it would really help if you can provide an intuition on why z_t still a martingale difference sequence although you are dealing with interconnected stochastic processes. Minor comments: - R_t is defined in line 121 and then again in line 136; you may want to merge the two definitions and state in line 121 that t denotes time. - Line 127, please mention the assumptions for the existence of the optimizer (such as unichain property) -Line 131: since both state and action space are finite, max notation may be more intuitive than sup =================== Post Rebuttal ======================== I'm satisfied with the author's response, and I decided to keep my score unchanged.


Review 2

Summary and Contributions: This paper develops a finite time analysis for the double Q-learning scheme with both synchronous and asynchronous implementations. It characterizes the convergence speed of double Q-learning to an epsilon accurate neighborhood, and bounds the difference between two interconnected stochastic process.

Strengths: The problem of reducing overestimation in RL is important and is timely. The paper is in general well written. The ideas are clearly explained.

Weaknesses: - As the paper states in the intro, double Q-learning was developed to address the overestimation problem of Q-learning. However, this cannot really be seen directly from the results in the paper. The explanation given in the paper suggests that double Q learning resolves the overestimation problem by achieving a fast convergence rate. While this is certainly related, is it the only key factor for this? It will be useful if the authors could provide more discussion on this. - While the reviewer understands that this is a theoretical work, it might be useful to provide numerical results to demonstrate the convergence and the block-wise bound. - tau^hat_1 appears to play an important role in the results. It would be useful to provide some intuition about this.

Correctness: Appears to be correct.

Clarity: Yes.

Relation to Prior Work: Yes.

Reproducibility: Yes

Additional Feedback: ===post rebuttal=== The reviewer thinks that the rebuttal is fine, and hopes to see the numerical results in the revision.


Review 3

Summary and Contributions: In this paper, the authors propose the first finite time analysis for both synchronous and asynchronous double Q-learning. They first analyzes the stochastic error propagation between two Q-estimators, then bound the error between one Q-estimator and the optimal Q function by conditioning on the error event associate with the previous analysis, and eventually bound the unconditional error between one Q-estimator and the optimum. The theoretical results are in line with the goal of double Q-learning, namely, reduce the number of aggressive move so as to avoid overestimation.

Strengths: 1. This is the first finite time analysis for double Q-learning, the theoretical result is exactly what we expect with when we are in the high accuracy regime (namely, \epsilon \ll 1-\gamma). 2. The technical contribution of the paper is sound, the authors introduce novel techniques to handle the inter-connected two random paths in double Q-learning.

Weaknesses: 1. In both Theorem 1 and Theorem 2, the discount factor \gamma are required to be greater than 1/3. I'm curious about why such the lower bound is imposed. 2. The primary focus of the paper is about the polynomial rate, can the current technique be applied to the case of linear rate?

Correctness: Although I didn't fully check the proof details in the Appendix (for example, I skip the proofs of Proposition 3 & 4), the main idea explained in the sketch proof is promising.

Clarity: The paper is reasonable well written, I appreciate the authors outline the sketch proof in the main paper with clear explanation of the high level idea.

Relation to Prior Work: Yes.

Reproducibility: Yes

Additional Feedback: =================== Post Rebuttal ======================== I'm satisfied with the author's response, and I decided to keep my score unchanged.


Review 4

Summary and Contributions: This submission investigated the theoretical properties of the double Q-learning, a popular algorithm in reinforcement learning. The authors derived the number of iterations needed for convergence with high probability, for both synchronous and asynchronous variants.

Strengths: The authors provided finite-time convergence analysis for the double Q-learning, which looks novel and interesting.

Weaknesses: The main results look similar to the ones in Even-Dar and Mansour (2003). The authors may need to provide more explanation on the connection and difference with this work, e.g., any new proof techniques used.

Correctness: This is a purely theoretical paper, and no experiments have been provided. I took a high-level check of the main results and they look to hold.

Clarity: This paper is in general well written but sometimes difficult to follow, as it's a pure theory paper and I am not an expert on the theoretical analysis of the relevant research.

Relation to Prior Work: Yes

Reproducibility: Yes

Additional Feedback: I am a bit confused about the definition between ``synchronous'' vs. ``asychronous'' in Section 2.2: The authors mentioned in L152-L153 that ``For synchronous double Q-learning (as shown in Algorithm 1), all the state-action pairs of the chosen Q-estimator are visited simultaneously at each iteration.'' However, in each iteration of Algorithm 1, only one state-action pair (s-a) was updated. Could you clarify the difference? === After rebuttal === Thank you for the response. Please add the clarification and connection with previous work into the revision.

[Author Response · NeurIPS 2020]

We thank the reviewers for the insightful reviews and valuable suggestions. We address the comments as follows.

**Reviewer #1:**

**Provide proof of Lemma 1:** The proof of Lemma 1 uses induction. We will add the proof to the supplementary.

**What is $R_{\max}$, a maximum value of the reward function $R$?** Yes, as defined in Section 2.1.

**Presentation issues of Algorithm 1:** Many thanks! Yes, Algorithm 1 is meant for the synchronous case. We will
clarify that lines 8 and 11 should be applied to all state action pairs. We will also specify the "converge" criterion in
Algorithm 1, which can be, for example, when $\sum_{i=t-N}^{t} \left\| Q_i^A - Q_{i-1}^A \right\|_\infty < \varepsilon$ for a given $N \geq 1$ and $\varepsilon > 0$.

**Discussion on the difficulties of analysis in the function approximation setting:** This is a very important yet
challenging problem. Even vanilla Q-learning with linear function approximation may not always converge and requires
strong assumptions/conditions to converge, because the contraction property of the Bellman operator no longer holds in
the function approximation setting. For double-Q algorithm, it is likely that neither of the two parameters (corresponding
to the two Q-functions) converges, and even if they both converge, it is unclear whether they converge to the same point.
Characterizing the conditions for these two interconnected stochastic processes to converge can be difficult.

**Discussion on exploration policy in asynchronous version and Assumption 1:** Yes, Assumption 1 is on a determin-
istic exploration policy for the simplicity of presentation of the key insights. The reason for $L \gg |S||A|$ is because
in practice often some state-action pairs are repeatedly visited before the first visit of some other state-action pairs.
$L = |S||A|$ can rarely happen in practice due to stochastic visits of state-action pairs. Assumption 1 can be replaced
by a high probability requirement. Our analysis can accommodate such a relaxation by additionally dealing with a
conditional probability event.

**In Step I of Part I of proof of Theorem 1, provide intuition on the martingale difference sequence (MDS) $z_t$ in**
**the modeling of the interconnected stochastic processes**: Indeed, interconnection of SAs introduces complication, as
reflected by the non-MDS error sequence $F_t$ (line 420 in supplementary). To handle the analysis of $F_t$, we decompose
$F_t = F_t - \mathbb{E}(F_t|\mathcal{F}_t) + \mathbb{E}(F_t|\mathcal{F}_t) := z_t + h_t$, where we define $z_t := F_t - \mathbb{E}(F_t|\mathcal{F}_t)$ and $h_t := \mathbb{E}(F_t|\mathcal{F}_t)$. In this way,
$z_t$ is constructed to be an MDS by subtracting the conditional mean of $F_t$ (a standard way to construct MDSs). The
complication of non-MDS nature of $F_t$ is captured by $h_t$, which we handle by exploiting a contraction-type property.

**Minor comments:** Many thanks for the suggestions. We will make the changes in the revision.

**Reviewer #2:**

**Discussion on double Q-learning resolving overestimation:** Our analysis of the interconnected SAs can also provide
some insights into how double Q-learning resolves overestimation. High-level speaking, the convergence of $\|Q^A - Q^*\|$
is obtained with high probability given the condition that $\|Q^A - Q^B\|$ can converge. This suggests that neither $Q^A$ nor
$Q^B$ can approach to $Q^*$ alone too aggressively, which implies mitigation of overestimation.

**Numerical results:** We have obtained some numerical results and will include them in the revision.

**Intuition on $\hat{\tau}_1$:** We design the length of blocks in a recursive way, so that the ending time of the first block $\hat{\tau}_1$
determines the ending times of all following blocks. Then in our analysis, $\hat{\tau}_1$ is determined to guarantee that the distance
between the two Q-functions can be bounded blockwisely with high probability.

**Reviewer #3:**

**Lower bound on $\gamma$ ( $\gamma > 1/3$):** In order for both $\{G_k\}$ and $\{D_k\}$ respectively serving as upper bounds on $\|Q^A - Q^B\|$
and $\|Q^A - Q^*\|$, we construct $G_k = \sigma D_k$ with $\sigma = \frac{1-\gamma}{2\gamma}$ in Proposition 2 (which may not be the only design to serve
the purpose). Then, since the convergence of $\|Q^A - Q^*\|$ is conditioned on the convergence of $\|Q^A - Q^B\|$, we further
need $D_k > G_k$, which requires $\gamma > 1/3$. This may not be an essential requirement, but is rather a technical assumption
due to the techniques we use.

**About linear learning rate:** Great question! We are currently working on extending our techniques to the case with
linear learning rate, and need to resolve some technical issues in order to obtain satisfactory results.

**Reviewer #4: Connection and difference with Even-Dar and Mansour (2003):** The connection lies in that our work
also used the blockwise analysis as in Even-Dar and Mansour (2003). The difference is that the analysis of double
Q-learning requires to handle two interconnected SAs, and the analysis of single SA in Even-Dar and Mansour (2003)
cannot be directly applied. Our technical novelty lies in designing the coupling relationship between two blockwise
upper bounds and dealing with conditional bounds.

**Definition between "synchronous" vs. "asynchronous":** Thanks for pointing this out. We will fix the statement of
the synchronous algorithm in the revision.

[Meta-Review · NeurIPS 2020]

The reviewers agreed that this paper provided a nice analysis of double Q-learning, thereby filling an open gap in the theoretical understanding of such algorithms, and unanimously recommended acceptance.